# PrimPol-dependent single-stranded gap formation mediates homologous recombination at bulky DNA adducts

Ann Liza Piberger[1✉], Akhil Bowry[1,5], Richard D. W. Kelly[1,5], Alexandra K. Walker[1], Daniel González-Acosta [2], Laura J. Bailey[3], Aidan J. Doherty [3], Juan Méndez [2], Joanna R. Morris [1], Helen E. Bryant[4] & Eva Petermann [1✉]

Stalled replication forks can be restarted and repaired by RAD51-mediated homologous recombination (HR), but HR can also perform post-replicative repair after bypass of the obstacle. Bulky DNA adducts are important replication-blocking lesions, but it is unknown whether they activate HR at stalled forks or behind ongoing forks. Using mainly BPDE-DNA adducts as model lesions, we show that HR induced by bulky adducts in mammalian cells predominantly occurs at post-replicative gaps formed by the DNA/RNA primase PrimPol. RAD51 recruitment under these conditions does not result from fork stalling, but rather occurs at gaps formed by PrimPol re-priming and resection by MRE11 and EXO1. In contrast, RAD51 loading at double-strand breaks does not require PrimPol. At bulky adducts, PrimPol promotes sister chromatid exchange and genetic recombination. Our data support that HR at bulky adducts in mammalian cells involves post-replicative gap repair and define a role for PrimPol in HR-mediated DNA damage tolerance.

[1] Institute of Cancer and Genomic Sciences, College of Medical and Dental Sciences, University of Birmingham, Birmingham B15 2TT, UK. [2] Molecular Oncology Program, Spanish National Cancer Research Centre, Madrid, Spain. [3] Genome Damage and Stability Centre, School of Life Sciences, University of Sussex, Falmer, Brighton BN1 9RQ, UK. [4] Department of Oncology & Metabolism, The Medical School, University of Sheffield, Sheffield S10 2RX, UK. [5] These authors contributed equally: Akhil Bowry, Richard D. W. Kelly. ✉email: A.L.Piberger@bham.ac.uk; E.Petermann@bham.ac.uk

A proper cell response to DNA damage during DNA replication is essential to maintain genome integrity and prevent cancer development. Environmental exposures frequently induce bulky DNA adducts, mutagenic, and carcinogenic DNA lesions that can pose strong physical obstacles to DNA replication forks. If not removed by nucleotide excision repair (NER), then the presence of these adducts on the DNA template inhibits the replicative polymerase, potentially leading to replication fork stalling. If stalled forks cannot be restarted, they can collapse into DNA double-strand breaks (DSBs), which are highly toxic and/or mutagenic. Fork collapse also activates homologous recombination (HR) for repair[1,2].

HR proteins, such as RAD51, are also involved in the remodeling and restart of stalled forks. We previously showed that replication inhibitors such as hydroxyurea (HU) induce at least two different RAD51-mediated HR pathways. Stalled forks recruit small amounts of RAD51 for restart, while HR repair of forks that have been collapsed into DSBs requires more extensive RAD51 loading and formation of RAD51 foci[3]. RAD51 functions at stalled forks have since then been linked to replication fork reversal[4], fork protection by preventing MRE11 resection[5], and promoting continuous replication through interaction with DNA Pol alpha[6]. Sources of bulky adducts, such as ultraviolet (UV) radiation or the environmental mutagen benzo[a]pyrene-diol-epoxide (BPDE), strongly induce RAD51 foci formation and HR, suggesting that these lesions stall and collapse forks[7–9]. However, bulky DNA lesions can also be bypassed using DNA damage tolerance pathways, which avoid fork stalling. Fork stalling can be avoided through re-priming, for example by PrimPol, a recently described RNA/DNA primase that exerts both DNA/RNA primase and DNA polymerase activity[10] and can re-prime after UV lesions[11–14]. Damaged DNA can be bypassed either through error-prone translesion synthesis (TLS) that is promoted by proliferating cell nuclear antigen (PCNA) mono-ubiquitination, or through an error-free damage tolerance pathway that is promoted by PCNA polyubiquitination and also uses recombination proteins.

This last pathway requires RAD51 and involves template switching to the undamaged sister chromatid. Mechanistically, template switching is still poorly understood. It has been proposed to operate either directly at the stalled fork, using fork reversal, or at post-replicative gaps behind the fork, while the fork itself continues by de novo re-priming[15]. In budding yeast, there is evidence that re-priming and post-replicative gap filling is preferred over fork reversal, and this pathway has been shown to involve a double Holliday junction-like intermediate[16,17]. Although it has been proposed since the 1970s that re-priming at DNA lesions can also lead to recombination in mammalian cells, fork reversal has been reported to be frequent in mammalian backgrounds[4], and may be preferred over re-priming.

Therefore, a longstanding and important mechanistic question is: where is RAD51 most important for replication bypass of bulky lesions, at the fork, or behind the fork? Does the observed HR activity at bulky lesions occur at stalled forks, collapsed forks or post-replicative gaps, and what are the molecular mechanisms involved?

To address these questions, we used BPDE-DNA lesions as models to investigate HR induction at bulky DNA adducts in human cells. Like UV lesions, BPDE adducts are highly relevant to human disease. They are induced by the common environmental carcinogen benzo[a]pyrene (B[a]P). B[a]P is one of the most potent known carcinogens and ubiquitous in barbequed meat as well as emissions from traffic, industry, stoves, and cigarette smoke[18,19]. BPDE adducts cause the specific mutation signature of smoking-induced lung cancer[20,21]. They are weak substrates for NER, and unrepaired adducts are ubiquitously detectable in the DNA of individuals[22,23]. BPDE adducts can be bypassed in both error-free and error-prone manner by TLS polymerases[24,25]. However, and in contrast to UV-induced cyclobutane pyrimidine dimers (CPDs), when replication encounters lower and more physiologically relevant levels of BPDE adducts, this predominantly induces HR[7,8]. BPDE adducts are thus highly recombinogenic and excellent model lesions for studying HR at bulky adducts (Fig. 1a).

Using low concentrations of BPDE that are non-toxic and recombinogenic, we report that RAD51 foci formation and HR activation in response to bulky DNA adducts can occur independently of replication fork stalling or -collapse. In these cases, RAD51 is recruited onto post-replicative single-stranded gaps that are generated by the re-priming activity of PrimPol, combined with resection. Our data support that HR activity at bulky adducts in mammalian cells can be associated with post-replicative repair and define a role for PrimPol in DNA damage tolerance.

## Results

### RAD51 foci form without fork stalling at bulky DNA adducts.
To investigate HR at bulky DNA adducts, U2OS cells were treated with BPDE for 20 min followed by release into fresh medium. After this treatment, BPDE lesions persist for more than 24 h due to slow repair[26]. We examined a range of concentrations from low non-cytotoxic (50 nM) to high and cytotoxic doses (500 nM and 1.65 µM) (Supplementary Fig. 1A). BPDE induced phospho-S139 histone H2AX (γH2AX) foci that co-localized with 5-Chlorodeoxyuridine (CldU) foci marking ongoing replication forks, supporting that the treatment was causing replication stress (Fig. 1b).

We then proceeded to quantify foci induction by BPDE. In all experiments, foci were quantified directly at the microscope, and images shown in the figures are for illustration. γH2AX foci were induced similarly across all BPDE concentrations and persisted for at least 24 h, consistent with inefficient repair (Fig. 1c). RAD51 foci analysis showed that HR was induced over several days after release from BPDE. In agreement with previous studies[7,8], the percentage of RAD51 foci positive cells were more strongly increased after release from 50 nM BPDE compared to higher BPDE doses (Fig. 1d, e). These observations were not simply due to cell cycle changes, as 50 nM BPDE had minimal effects on cell cycle distribution, and 500 nM BPDE induced an S phase rather than a G1 arrest (Supplementary Fig. 1B, C). Furthermore, the numbers of RAD51 foci per positive cell were also strongly increased after 50 nM BPDE (Supplementary Fig. 1D).

We next determined the impact of BPDE treatment on replication fork slowing and -stalling. This first required developing optimized CldU and IdU dual pulse-labeling protocols for DNA fiber analysis. We initially added BPDE with the second (IdU) label to measure the immediate effect of BPDE on replication fork progression. Only very high BPDE concentrations (1.65 µM) strongly slowed replication speeds (Supplementary Fig. 1E). To measure replication fork stalling, we labeled U2OS cells with CldU before, during, and for 30 min after the BPDE incubation to detect forks that stalled either during or after the treatment. Cells were then washed and labeled with IdU to detect any ongoing forks for a further 20 min (Fig. 1f). Stalled forks were defined as CldU tracks not followed by an IdU track. Based on our previous measurements[26,27], we calculate that a 20 min treatment with 50 nM BPDE would induce 150 adducts/$10^8$ base pairs. If an average replication fork replicates $6 \times 10^4$ base pairs[28], about 10% of forks would likely encounter a lesion. However, while higher BPDE concentrations increased fork

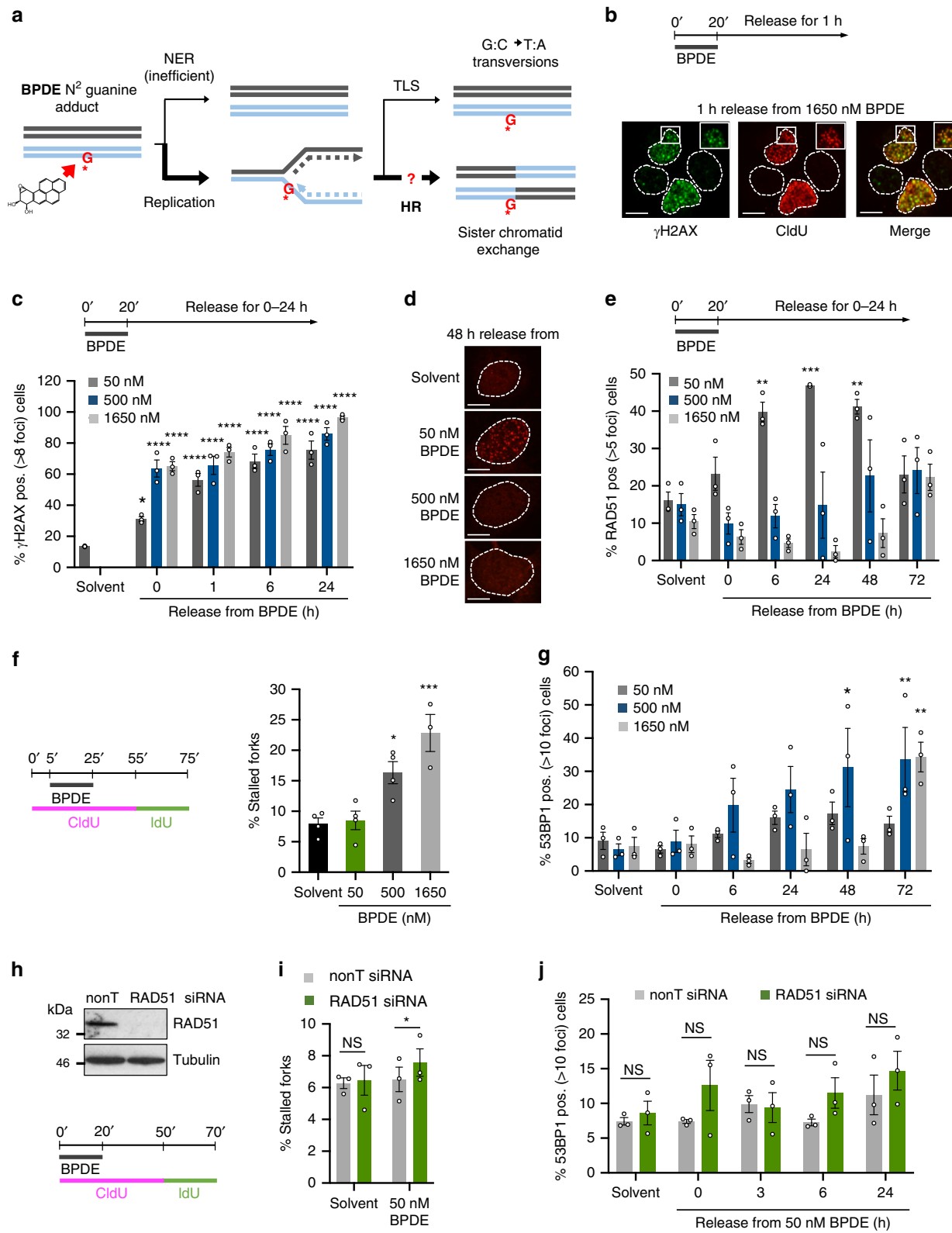

stalling, 50 nM BPDE did not lead to appreciable increase in stalled forks (Fig. 1f). Similar results were also obtained using fork asymmetry as a readout for fork stalling (Supplementary Fig. 1F). This suggested that replication forks could continue progression on the damaged template under these conditions.

We previously reported that RAD51 recruitment to stalled forks does not lead to RAD51 foci formation, which instead

requires fork collapse into DSBs[3]. This raised the question whether treatment with 50 nM BPDE could cause DSB formation. Analysis of 53BP1 foci, a DSB marker, and physical detection of DSBs using pulse-field gel electrophoresis (PFGE) suggested that 50 nM BPDE induced few DSBs, while 500 nM and 1.65 μM BPDE did induce DSB formation (Fig. 1g, Supplementary Fig. 1G, H). Interestingly, highly cytotoxic treatment with 1.65 μM BPDE

**Fig. 1 Bulky DNA adducts induce RAD51 foci formation in absence of replication fork stalling or -collapse. a** Schematic of DNA repair and damage tolerance pathways induced by low doses of BPDE, which forms bulky adducts on guanine. NER: nucleotide excision repair; TLS: translesion synthesis; HR: homologous recombination. **b** Co-localization of γH2AX (green) and replication forks (CldU, red) in U2OS cells after 1 h release from 1650 nM BPDE treatment for 20 min. Images are representative of $n = 2$. Scale bars: 10 μm. **c** Percentages of U2OS cells with >8 γH2AX foci after release from increasing concentrations of BPDE. $n = 3$ **d** BPDE-induced RAD51 foci. Images are representative of $n = 3$. Scale bars: 10 μm. **e** Percentages of U2OS cells with >5 RAD51 foci after release from increasing concentrations of BPDE. $n = 3$. **f** Percentages of stalled forks after release from BPDE. U2OS cells were pulse-labeled with CldU, treated with BPDE during the CldU pulse, and released into IdU. $n = 4$ (1650 nM $n = 3$). **g** Percentages of cells with >10 53BP1 foci after release from increasing concentrations of BPDE. $n = 3$. **h** Top: Protein levels of RAD51 and α-Tubulin (loading control) in U2OS cells after 48 h depletion with RAD51 or non-targeting (nonT) siRNA. Images are representative of $n = 2$. Bottom: Strategy for DNA fiber labeling. **i** Percentages of stalled forks after release from 50 nM BPDE as shown in (**h**) in cells treated with nonT or RAD51 siRNA for 48 h. $n = 3$. **j** Percentages of control- or RAD51-depleted U2OS cells with >10 53BP1 foci after release from 50 nM BPDE. $n = 3$. Source data are provided as a Source Data file. The means and SEM (bars) of at least three independent experiments are shown. Asterisks indicate $p$-values compared to solvent unless indicated otherwise (one-way ANOVA and Dunnett's test for C, E, F, G; one-sided student's t-test for I, J, *$p < 0.05$, **$p < 0.01$, ***$p < 0.001$).

may suppress the DSB response downstream of γH2AX, as the DSBs detected by PFGE did not coincide with RAD51 or 53BP1 foci induction under these conditions (Fig. 1e, g, Supplementary Fig. 1H). Taken together, these data provided no evidence that RAD51 foci induced by 50 nM BPDE correlate with DSB formation. In contrast, RAD51 foci induced by 500 nM BPDE correlated well with DSB formation (Fig. 1e, g). This posed the question as to the mechanism and role of RAD51 foci formation specifically after low dose BPDE treatment.

RAD51 is required to promote fork restart in response to HU treatment[3]. In contrast, siRNA depletion of RAD51 did not increase levels of fork stalling after release from low (50 nM) or high (1.65 μM) BPDE concentrations (Fig. 1h, i, Supplementary Fig. 2A). RAD51 can also promote fork slowing or -stalling at lesions induced by UV or methyl methanesulfonate (MMS)[29,30]. However, we found no evidence that RAD51 promotes fork slowing on BPDE-damaged templates (Supplementary Fig. 2B). Furthermore, RAD51 is required for the repair of collapsed forks and prevents DSB accumulation in response to HU[3]. However, RAD51 depletion had no major effect on the levels of 53BP1 foci induced by either 50 nM or 500 nM BPDE (Fig. 1j, Supplementary Fig. 2C). These experiments provided no evidence that RAD51 is required to prevent fork collapse, or repair collapsed forks in response to BPDE lesions.

**PrimPol re-priming promotes gap formation at bulky adducts.** RAD51 loading to initiate HR requires the formation of long stretches of ssDNA that are initially coated with RPA, which is then exchanged for RAD51. RPA foci analysis showed that ssDNA was rapidly formed even after release from 50 nM BPDE, in absence of fork stalling or DSB formation (Supplementary Fig. 2D, E). RPA foci were resolved between 24 and 48 h after release from BPDE (Fig. 2a). γH2AX foci were also resolved between 24 and 48 h after release from BPDE, suggesting that they mostly mark ssDNA (Supplementary Fig. 2F). RAD51 depletion impaired the resolution of BPDE-induced RPA foci, suggesting that RAD51 was involved in the repair of BPDE-induced ssDNA (Fig. 2a).

To specifically test whether ssDNA gaps were formed at BPDE lesions behind ongoing forks, we used the S1 endonuclease (S1)-modified fiber assay[31]. S1-dependent DNA cleavage of ssDNA gaps or nicks allows to specifically detect these lesions in newly replicated DNA (Fig. 2b). After DNA fiber labeling, nuclei are permeabilized and treated with recombinant S1 nuclease before DNA fiber spreading and staining. The IdU pulse is extended to 50 min in order to capture more events, even though this leads to CldU/IdU ratios that are skewed upwards compared to what would be expected based on the labeling times (Fig. 2c, Supplementary Fig. 3A, B). However, if ssDNA gaps are present in nascent DNA, the S1-induced DSBs cause further shortening of

the IdU-labeled fibers, thus increasing CldU/IdU ratios compared to mock-treated samples. Release from 50 nM BPDE, but not from the solvent, resulted in increased CldU/IdU ratios in S1 treated- compared to mock-treated samples, supporting that ssDNA gaps were generated behind the fork during replication of DNA containing BPDE lesions (Fig. 2c). Taken together, these data suggested that RAD51 is involved in post-replicative repair and that the RAD51 foci nucleation site might be at regions of ssDNA away from the ongoing fork. We hypothesized this ssDNA might form at post-replicative gaps through re-priming of DNA synthesis downstream of the bulky lesion (Fig. 2d).

We, therefore, investigated a potential role for re-priming in ssDNA gap formation. The RNA/DNA primase PrimPol was recently reported to re-prime after UV lesions[11–14] and was therefore a good candidate for promoting ssDNA gap formation (Fig. 2d). We used siRNA[12] to deplete PrimPol in U2OS cells (Fig. 2e, Supplementary Fig. 3C). We first tested whether PrimPol re-priming was responsible for the continued fork progression and lack of fork stalling after treatment with 50 nM BPDE. Indeed, we observed that 50 nM BPDE induced fork stalling specifically when PrimPol was depleted (Fig. 2f). Furthermore, PrimPol depletion prevented ssDNA gap induction behind the fork as measured by S1 nuclease fiber assay (Fig. 2g, Supplementary Fig. 3D–F). One caveat is that we did not perform the S1 nuclease fiber assay also under conditions of inhibiting NER and base excision repair, which both could induce ssDNA gaps and contribute to S1 fiber shortening after treatment with BPDE. However, excision repair-induced gaps would occur independently of PrimPol, thus not affecting the overall impact of PrimPol depletion.

PrimPol exerts both DNA/RNA primase and DNA polymerase activity[10]. To test whether the PrimPol primase activity is specifically required for continued replication fork progression in presence of BPDE, we performed rescue experiments by ectopically expressing wild type (WT) and primase-dead (CH) mutant versions of PrimPol[11,32] (Fig. 2h). These experiments were performed both using a U2OS cell line carrying an inducible shRNA that would not target the ectopic constructs[7] (Fig. 2i), as well as siRNA depletion combined with re-expression of siRNA-resistant WT or CH mutant PrimPol variants (Supplementary Fig. 3G). Only WT PrimPol, but not CH PrimPol, could rescue BPDE-induced fork stalling, supporting that re-priming prevents BPDE-induced fork stalling (Fig. 2j, Supplementary Fig. 3H, I).

**PrimPol re-priming and end processing promote RAD51 loading.** We next investigated whether RAD51 foci formation itself depended on PrimPol-mediated re-priming and ssDNA gap formation. PrimPol depletion in U2OS cells specifically suppressed BPDE-induced RAD51 foci formation but did not suppress spontaneous RAD51 foci (Fig. 3a, b). BrdU incorporation

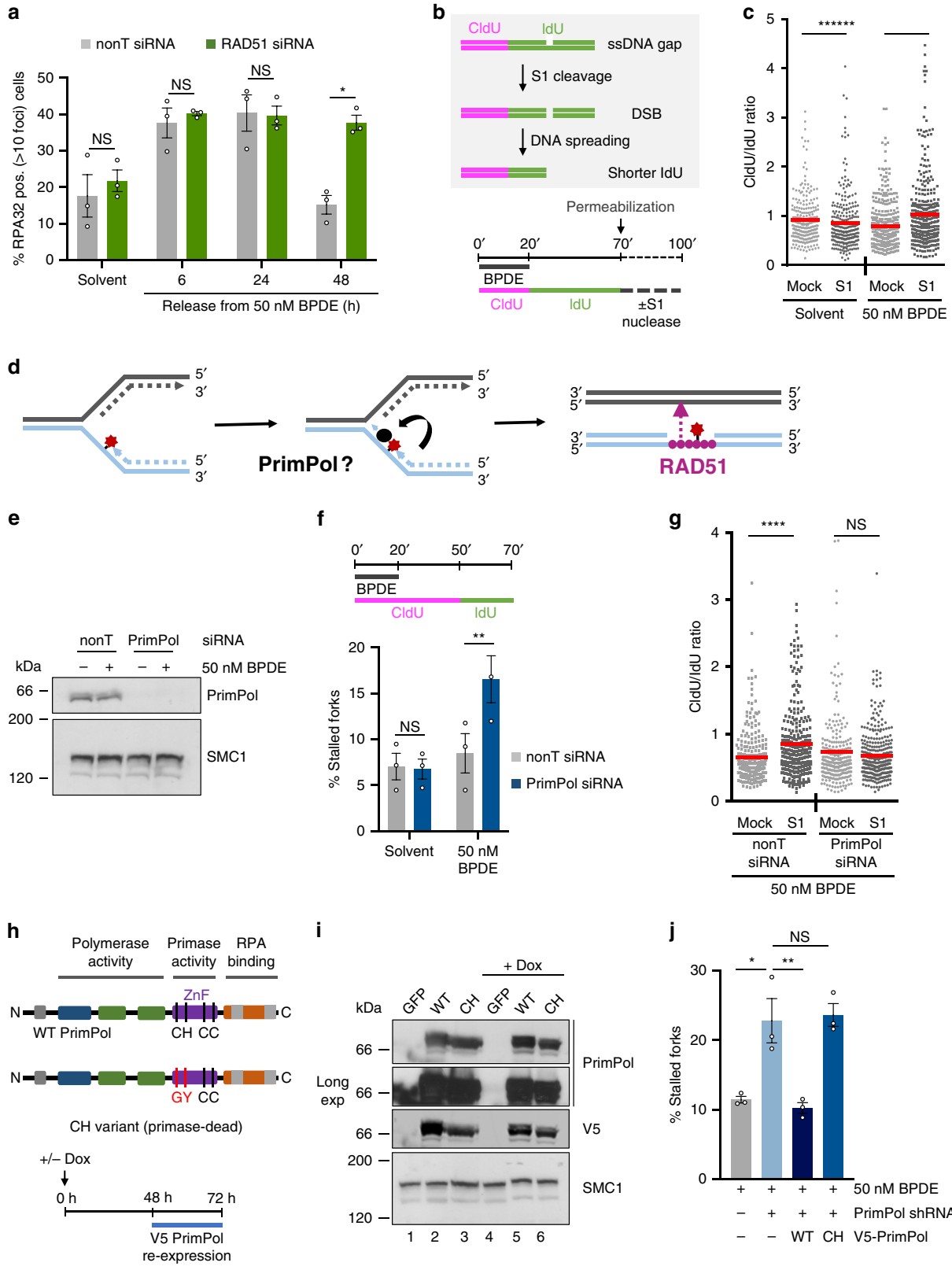

confirmed that this was not due to cell cycle changes or inhibition of DNA replication after PrimPol depletion (Fig. 3c, Supplementary Fig. 4A). Ongoing *PRIMPOL* mRNA depletion over the time course was confirmed using qRT-PCR (Supplementary Fig. 4B). Additionally, impaired RAD51 foci formation was also observed in PrimPol-depleted epithelial lung carcinoma A549

and normal lung fibroblast MRC5 cells, and in *PRIMPOL* knock-out MRC5 cells (Supplementary Fig. 4C–E).

We next performed rescue experiments to test whether PrimPol primase activity is needed for RAD51 foci formation in presence of BPDE. RAD51 foci formation in PrimPol-depleted cells could be rescued by ectopic expression of PrimPol WT, but

**Fig. 2 Post-replicative single-stranded DNA gaps are generated at bulky adducts through re-priming by PrimPol. a** Percentages of nonT- or RAD51-siRNA treated U2OS cells with >10 RPA foci after release from BPDE. Release from solvent was 3 h. n = 3. **b** Schematic of the S1 endonuclease-modified DNA fiber assay and strategy for DNA fiber labeling. **c** CldU/IdU ratios after S1-modified DNA fiber assay in cells treated with solvent or 50 nM BPDE as in (**b**). S1: S1 endonuclease. Lines represent mean. Data from 3 repeats. **d** Schematic of RAD51-mediated ssDNA gap repair after re-priming at replication-blocking lesions. **e** Protein levels of PrimPol and SMC1 (loading control) after 6 h release from 50 nM BPDE or solvent (-) in U2OS cells after 48 h depletion with PrimPol or nonT siRNA. kDa: kiloDalton. Representative of n = 3. **f** Quantification of stalled forks after release from 50 nM BPDE in presence of nonT or PrimPol siRNA. n = 3. **g** CldU/IdU ratios after S1-modified DNA fiber assay in cells treated with 50 nM BPDE as in (**b**) after 48 h of nonT or PrimPol siRNA. Lines represent mean. Data from 3 repeats. **h** Top: Protein domain structure of wild type (WT) and primase-dead (CH) PrimPol variants. Bottom: Protocol for dox-inducible shRNA depletion of endogenous PrimPol and ectopic expression of V5-tagged PrimPol variants. **i** Protein levels of ectopic V5-tagged WT or CH PrimPol, endogenous PrimPol (long exposure, lane 1 and 4), and SMC1 (loading control) in U2OS cells after treatment as indicated above. Representative of n = 3 (-Dox:WT and -Dox:CH n = 2). **j** Quantification of stalled forks after release from 50 nM BPDE with or without PrimPol shRNA and expression constructs encoding GFP (-) or WT or CH PrimPol. Fiber labeling was as in (**f**) n = 3. Source data are provided as a Source Data file. The means and SEM (bars) of at least three independent experiments are shown. Asterisks indicate p-values compared to solvent unless indicated otherwise (one-sided student's t-test for A, F, one-sided Mann–Whitney for C, G; one-way ANOVA for J, *p < 0.05, **p < 0.01, ***p < 0.001, ****p < 0.0001).

not the primase-dead PrimPol CH mutant, again suggesting that specifically the re-priming activity is required for RAD51 foci formation (Fig. 3d). Similar results were obtained using siRNA depletion combined with the expression of siRNA-resistant PrimPol variants (Supplementary Fig. 4F).

*In vitro* experiments suggest that PrimPol re-primes around 14 nucleotides downstream of DNA lesions[33]. Such short ssDNA gaps would need further resection by nucleases such as MRE11 and Exonuclease 1 (EXO1) to allow for RAD51 foci formation (Fig. 4a). To investigate the resection of these gaps, we used single-molecule analysis of resection tracks (SMART)[34]. Cells were treated with BrdU for 48 h to BrdU-label all DNA (Fig. 4b). DNA fiber spreading followed by BrdU immunostaining without acid denaturation was then used to visualize stretches of ssDNA (Fig. 4c). BPDE treatment clearly induced ssDNA tracks, which required both PrimPol and EXO1 (Fig. 4c, d). In contrast, ssDNA tracks at DSBs induced by etoposide were considerably longer than BPDE-induced tracks and required EXO1, but not PrimPol (Supplementary Fig. 5A). This supports that BPDE-induced ssDNA tracks were not due to resected DSBs. To further investigate the contribution of resection activities to BPDE-induced ssDNA track formation, we used PFM01, a chemical inhibitor of the endonuclease activity of MRE11 (Fig. 4e). PFM01 treatment also reduced the lengths of the ssDNA tracks (Fig. 4e, f), suggesting that initial short gaps are further extended by resection through MRE11 and EXO1. MRE11 and EXO1 activities may be additive in promoting the resection of the ssDNA gaps, which we have not tested.

We then investigated the impact of resection on RAD51 foci formation, using chemical inhibitors of the exo- and endonuclease activities of MRE11 (mirin and PFM01, respectively) as well as siRNA depletion of EXO1. MRE11 activity and EXO1 were both required for BPDE-induced RAD51 foci formation, supporting that PrimPol-mediated HR initiation involves resection (Fig. 4g, h).

**PrimPol also promotes RAD51 loading at UV and 4-NQO adducts.** To investigate the relevance of PrimPol re-priming for HR initiation at other bulky lesions, we used UV-C irradiation to induce CPD and 6-4 photoproducts and measured the impact on ssDNA gap formation in the presence and absence of PrimPol (Fig. 5a). UV-C irradiation induced large amounts of S1 endonuclease-sensitive sites, as the lesion density would be higher than after 50 nM BPDE at about 150 adducts/$10^6$ base pairs[35]. While PrimPol depletion induced some S1-sensitive sites in both mock- and UV-C-irradiated samples, the UV-specific induction of S1-sensitive sites was reduced after PrimPol depletion (Fig. 5b, Supplementary Fig. 6A, B). Furthermore, PrimPol was required

for efficient RAD51 foci formation in response to bulky lesions induced by UV-C, and at bulky DNA adducts induced by the chemical carcinogen 4-nitroquinolie 1-oxide (4-NQO), which induces bulky adducts on purines (Fig. 5c, d). PrimPol requirement for RAD51 formation after UV-C was overall smaller than for BPDE and 4-NQO. This may be partly due to recombination also being initiated by NER activity at UV lesions[36]. Taken together, these data suggest that PrimPol-mediated re-priming generates ssDNA gaps for RAD51 loading at bulky DNA adducts more widely.

We then investigated whether PrimPol is required for RAD51 foci formation under conditions where HR is initiated at DSBs. We used treatments with 2 mM HU for 24 h, or 2 Gy of ionizing radiation (IR), which both induce DSBs either via fork collapse or directly[3]. PrimPol was not required for RAD51 loading in response to DSB-inducing treatments (Fig. 5e, f). Similarly, treatment with 500 nM BPDE induces predominantly DSB-associated HR after fork stalling (Fig. 1e–g), and PrimPol was not required for RAD51 loading in response to 500 nM BPDE (Supplementary Fig. 6C). PrimPol re-priming still occurred at 500 nM BPDE (Supplementary Fig. 6D, E), indicating that PrimPol-mediated ssDNA gaps can be channeled into other gap-filling pathways such as TLS. This observation is consistent with reports that high BPDE damage loads favor TLS over HR[7,8]. The lack of a role for PrimPol in DSB-induced recombination agrees with the standard model of RAD51 loading onto resected DSB ends, which does not require any re-priming.

**PrimPol promotes HR and sister chromatid exchanges.** Finally, we investigated whether PrimPol-mediated re-priming is required to support the activation of the complete HR pathway leading to genetic recombination. Both gene conversion (GC) and sister chromatid exchanges (SCEs) have been proposed as possible products of recombination by template switching[37,38]. SCE formation is usually attributed to collapsed fork repair, but according to some models, SCEs could also arise from post-replicative template switching[38]. We first analyzed BPDE-induced recombination frequencies using the reporter cell line SW480SN.3, which harbors the SCneo reporter construct that can detect recombination by GC or SCE[39]. 50 nM BPDE induced measurable recombination in this reporter, and this was reduced after PrimPol depletion (Fig. 6a, Supplementary Fig. 7A). In contrast, there was no evidence that PrimPol depletion affected colony survival or percentage of sub-G1 population after BPDE treatment, and the impact of RAD51 depletion on survival was also very small (Fig. 6b, Supplementary Fig. 7B). PrimPol depletion did also not increase DSB formation after BPDE

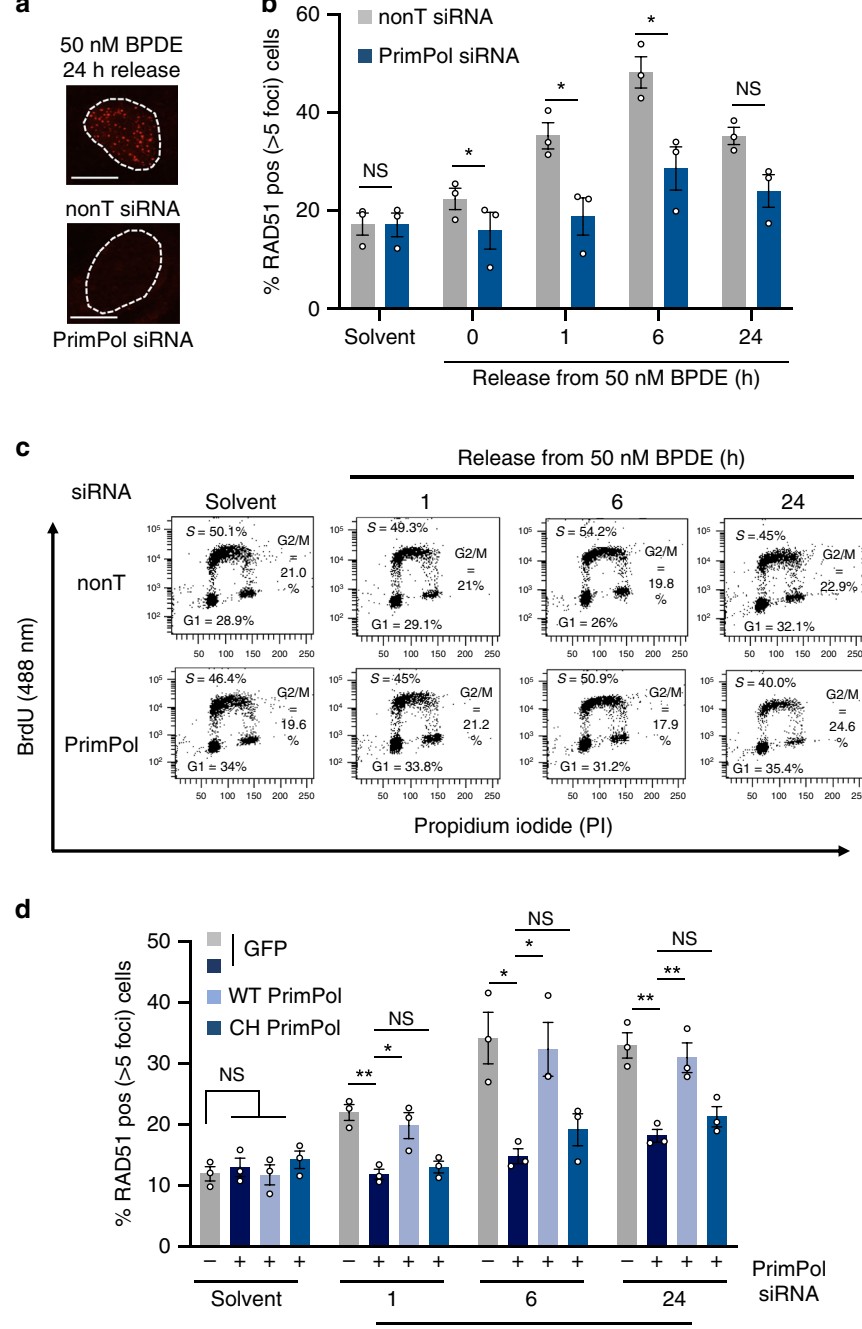

**Fig. 3 PrimPol-mediated re-priming promotes RAD51 loading in response to bulky DNA adducts. a** RAD51 foci after 24 h release from 50 nM BPDE in the presence of nonT or PrimPol siRNA. Images are representative of $n = 3$. Scale bars: 10 μm. **b** Percentages of U2OS cells with >5 RAD51 foci after release from 50 nM BPDE, in presence of nonT or PrimPol siRNA. $n = 3$. **c** Representative cell cycle distributions after release from solvent (1 h) or 50 nM BPDE in presence of nonT or PrimPol siRNA. Cells were incubated with BrdU for 30 min to label replicating cells prior to flow cytometry. Percentages are means from three independent experiments as shown in Figure S4A. **d** Percentages of cells with >5 RAD51 foci after release from 50 nM BPDE, with or without PrimPol shRNA depletion and expression constructs encoding GFP or WT or CH PrimPol. The release from the solvent was 1 h. $n = 3$. Source data are provided as a Source Data file. The means and SEM (bars) of at least three independent experiments are shown. Asterisks indicate $p$-values (one-sided student's t-test for B or one-way ANOVA for D, $*p < 0.05$, $**p < 0.01$).

treatment (Supplementary Fig. 7C). This is in line with previous findings that PrimPol-depleted human cells are not very sensitive to DNA damage induced by UV, suggesting that other pathways such as TLS or fork regression can compensate[12,40]. It also suggests that the PrimPol-mediated pathway might predominantly impact on BPDE-induced mutagenesis and/or the level of genomic rearrangements, rather than cytotoxicity. We further used microscopy to investigate SCE induction as a read-out for crossover resolution of recombination events. 50 nM BPDE and 50 nM 4-NQO both clearly induced SCE formation (Fig. 6c–e). In support of this, PrimPol depletion prevented BPDE- and 4-NQO-induced SCE formation, while it did not influence the spontaneous induction of SCEs observed under undamaged conditions (Fig. 6d, e).

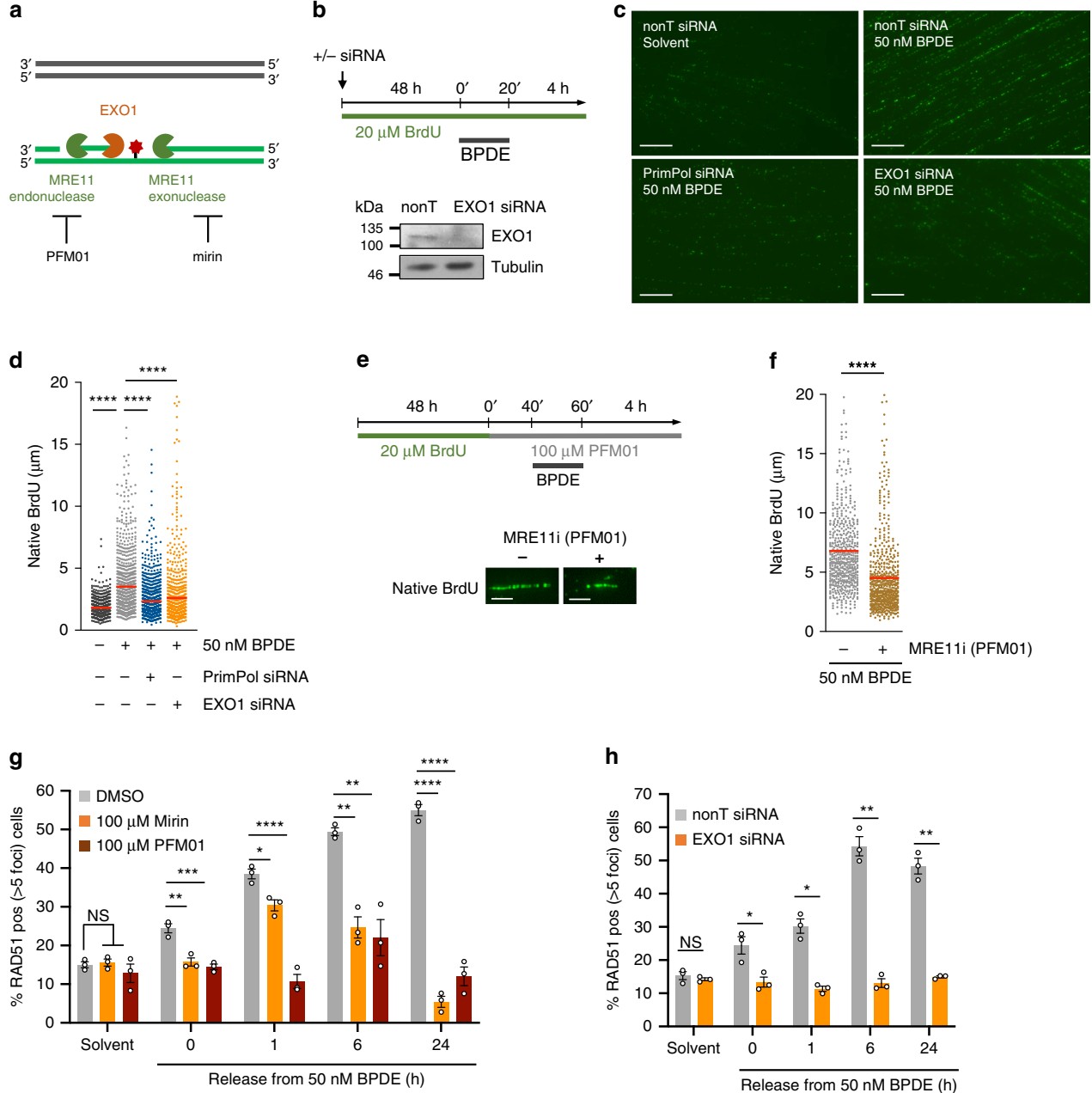

**Fig. 4 RAD51 loading at PrimPol-dependent single-stranded DNA gaps requires DNA end resection. a** Schematic of MRE11 and EXO1 nuclease activities and targeting by MRE11 inhibitors PFM01 and mirin. **b** Top: Strategy for SMART analysis of single-stranded DNA quantifications. Bottom: Protein levels of EXO1 and α-Tubulin (loading control) in U2OS cells after 48 h depletion with EXO1 or nonT siRNA. Images are from $n = 1$. **c** SMART analysis of single-stranded DNA after release from 20 min of 50 nM BPDE or solvent (-) for 4 h, in presence of nonT (-), PrimPol, or EXO1 siRNA. Images are representative of $n = 3$. Scale bars: 10 μm. **d** Quantification of SMART analysis after BPDE as in (**c**). Lines represent mean. Data from 3 repeats. **e** Strategy for SMART analysis with MRE11 inhibitor (MRE11i) PFM01 and representative images of native BrdU tracts. Scale bars: 5 μm. **f** SMART analysis of single-stranded DNA after release from 20 min of 50 nM BPDE for 4 h, in the presence or absence of MRE11 inhibitor PFM01. Lines represent mean. Data from 3 repeats. **g** Percentages of cells with >5 RAD51 foci after release from 50 nM BPDE, in the presence or absence of MRE11 inhibitors mirin or PFM01 as in (**e**). $n = 3$. **h** Percentages of cells with >5 RAD51 foci after release from 50 nM BPDE, in the presence of nonT or EXO1 siRNA for 48 h. $n = 3$ Source data are provided as a Source Data file. The means and SEM (bars) of at least three independent experiments are shown. Asterisks indicate p-values (one-way ANOVA for D, G, one-sided Mann–Whitney for F, one-sided student's t-test for H, $*p < 0.05$, $**p < 0.01$, $***p < 0.001$, $****p < 0.0001$).

## Discussion

We have shown that during DNA replication of bulky DNA adducts in mammalian cells, HR activation can occur in absence of stalled or collapsed replication forks at single-stranded gaps that are generated by the re-priming activity of PrimPol (Fig. 6f). This pathway has both similarities and differences to DSB repair.

Previous work has suggested roles for re-priming and PrimPol at UV and cisplatin lesions[9,11–14,41]. It is therefore to be expected that re-priming also occurs at other bulky lesions. However, our findings address long-standing questions by showing that at bulky DNA adducts, RAD51 foci formation and sister chromatid exchange that have been traditionally connected with replication fork collapse and DSB repair, are associated with the repair of

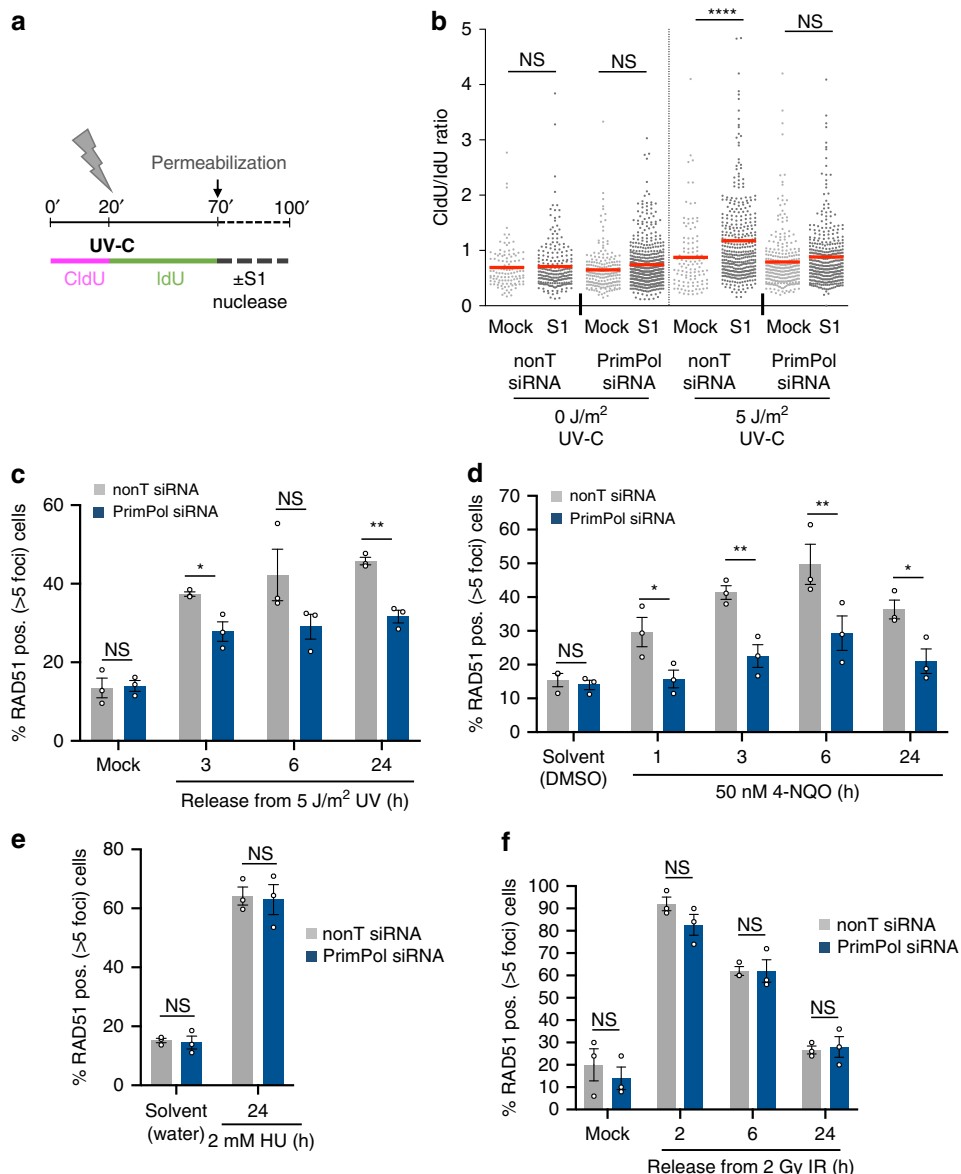

**Fig. 5 PrimPol promotes RAD51 foci formation specifically in response to bulky adducts. a** Strategy for the S1-modified DNA fiber assay with ultraviolet (UV-C) irradiation. **b** CldU/IdU ratios after S1-modified DNA fiber assay in cells treated with 5 J/m² UV-C or mock-treated as in (**a**), in the presence of nonT or PrimPol siRNA. Lines represent mean. Data from 4 repeats. **c** Percentages of cells with >5 RAD51 foci after release from 5 J/m² UV, in presence of nonT or PrimPol siRNA. Release from mock-irradiation was 3 h. $n = 3$. **d** Percentages of cells with >5 RAD51 foci after treatment with 50 nM 4-nitroquinolie 1-oxide (4-NQO), in presence of nonT or PrimPol siRNA. The release from DMSO was 1 h. $n = 3$. **e** Percentages of cells with >5 RAD51 foci after treatment with 2 mM hydroxyurea (HU) for 24 h, in presence of nonT or PrimPol siRNA. $n = 3$. **f** Percentages of EdU positive cells with >10 RAD51 foci after release from 2 Gy ionizing radiation (IR), in the presence of nonT or PrimPol siRNA. Release from mock-irradiation was 2 h. $n = 3$. Source data are provided as a Source Data file. The means and SEM (bars) of at least three independent experiments are shown. Asterisks indicate p-values (One-way ANOVA for B; one-sided student's $t$-test for C-F, $*p < 0.05$, $**p < 0.01$, $***p < 0.001$, $****p < 0.0001$).

post-replicative gaps. Furthermore, these post-replicative gaps are produced by PrimPol, shedding light on the function of PrimPol during DNA damage tolerance.

Most DNA damaging or replication blocking treatments cause fork slowing and/or stalling as well as fork reversal[4]. The previously described functions of RAD51 at such forks are therefore likely to be active during most forms of replication stress. However, we suggest that in addition to fork reversal, fork restart and the repair of collapsed forks, HR at post-replicative gaps constitutes an additional important function of RAD51 in response to replication blocks. There could be competition between several of these pathways. It was recently reported that PrimPol repriming competes with replication fork reversal in response to

cisplatin lesions[41]. This work also showed that PrimPol mRNA and protein levels increase over 24 h after cisplatin and UV treatment, thus favoring repriming over fork reversal as an adaptive mechanism[41]. However, this response was specific for BRCA1-deficient backgrounds[41], and our qRT-PCR data show no increase in *PRIMPOL* mRNA expression over 24 h after BPDE in BRCA1-proficient U2OS cells (Supplementary Fig. 4B).

Why are these post-replicative gaps resected and channeled into HR rather than simply filled by TLS? We speculate that HR and TLS-dependent gap-filling could compete at ssDNA gaps. Gap resection might prevent TLS and promote HR, but TLS might be preferred over resection under high damage load. It will be important to decipher whether and how PrimPol re-priming

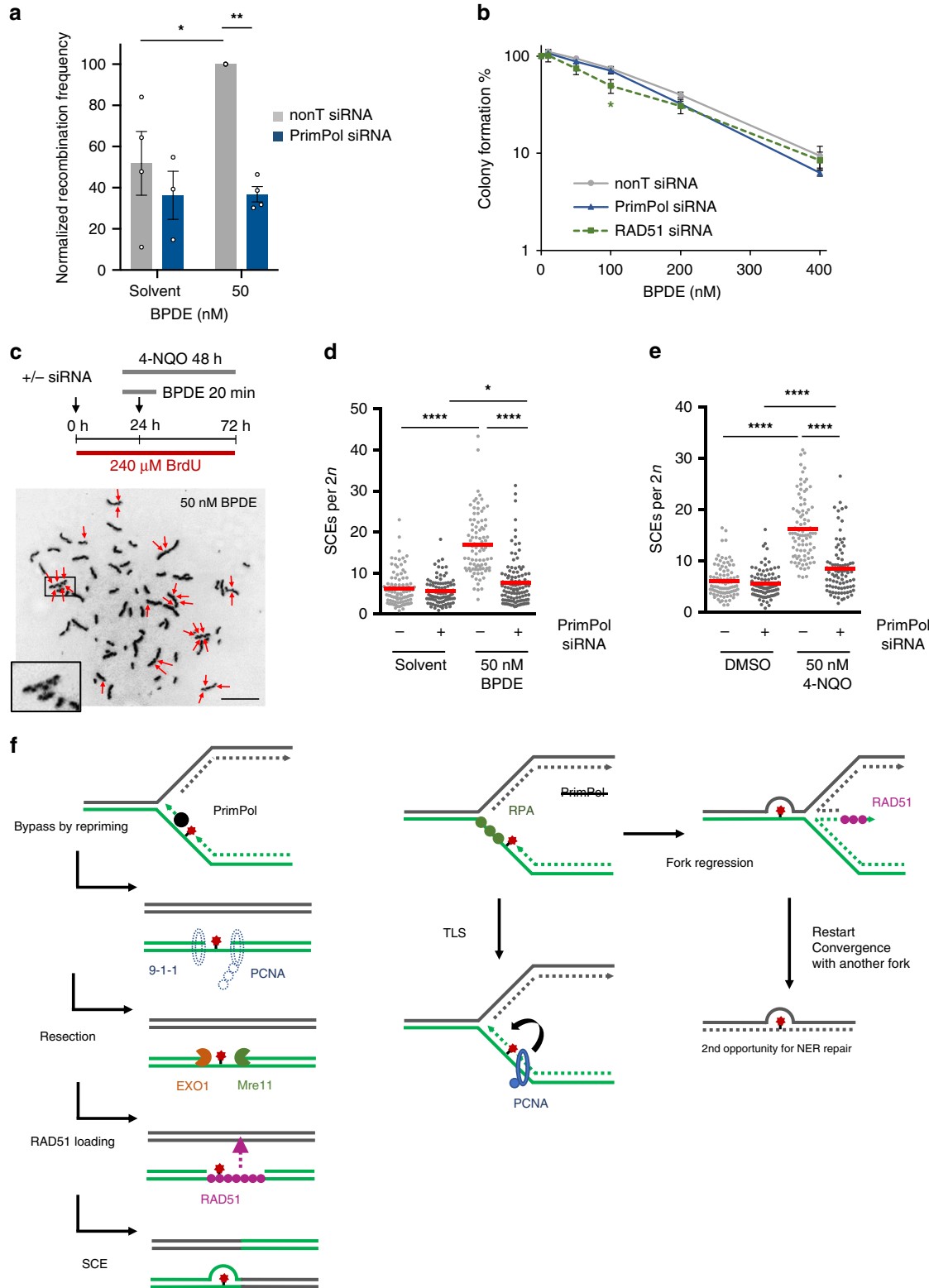

and the presence of the bulky adduct influence the pathway decision between TLS and template switching. Future work could involve examining the impact of PrimPol on PCNA ubiquitylation.

Our data agree with recent work in budding yeast, reporting that checkpoint activation by fork-stalling lesions also occurs at EXO1-resected post-replicative gaps rather than directly at the fork[42]. Furthermore, EXO1 resection is downstream of PCNA

polyubiquitylation that promotes template switching. A model was proposed in which resection of the gap is stimulated by polyubiquitylated PCNA stuck on the 3′ (upstream) junction and by the 9-1-1 checkpoint clamp loaded onto the 5′ (downstream) junction[43] (Fig. 6f). The length of BPDE-induced gaps will require further investigation to determine whether PrimPol can re-prime far away from the lesion, and/or whether long-range resection takes place.

**Fig. 6 PrimPol is required for homologous recombination induced by bulky adducts. a** Relative recombination frequencies in SW480SN.3 cells induced by BPDE. Cells were treated with 50 nM BPDE for 20 min in presence of nonT or PrimPol siRNA. n = 4 (PrimPol siRNA solvent n = 3). **b** Colony survival assay of U2OS cells after 20 min treatment with BPDE at the indicated concentrations in presence of nonT, PrimPol or RAD51 siRNA. n = 4 (RAD51 siRNA n = 3). **c** Strategy for sister chromatid exchange (SCE) detection and representative picture of metaphase spreads in response to 50 nM BPDE or 4-NQO. Scale bar: 10 μm. **d** SCE formation in U2OS cells after treatment with 50 nM BPDE or solvent in the presence of nonT or PrimPol siRNA. Lines represent mean. Data from 3 repeats. **e** SCE formation in U2OS cells after treatment with 50 nM 4-NQO or DMSO in the presence of nonT or PrimPol siRNA. Lines represent mean. Data from 3 repeats. **f** Model of recombination at bulky DNA adducts. Left: In presence of PrimPol, re-priming creates gaps that are resected for RAD51 foci formation and recombination repair. Potential TLS events at gaps and recombination not resulting in sister chromatid exchanges have been omitted for simplicity. Right: In absence of PrimPol, bulky DNA adducts are either bypassed by TLS or alternatively, stalled forks are stabilized by fork regression and potential rescue upon convergence with another fork. Source data are provided as a Source Data file. The means and SEM (bars) of at least three independent experiments are shown. Asterisks indicate p-values (one-way ANOVA, *p < 0.05, **p < 0.01, ***p < 0.001, ****p < 0.0001).

With the observed roles for resection and RAD51 foci formation, HR at bulky adducts seems to resemble DSB repair more than the known HR functions at stalled forks. It has been speculated pro-recombination activities could be specifically recruited to the post-replicative environment at the gap[15]. These may promote RAD51 foci formation, and long-range resection may support a more accurate homology search. In contrast to stalled replication forks, RAD51 foci formation might be suppressed by anti-recombinogenic helicase activities[15] and the possibilities for remodeling, e.g., by fork reversal (Fig. 6f).

Another unanswered question is the contribution of re-priming by DNA polymerase α (Pol α) to HR. Pol α is solely responsible for gap formation in yeast, which lacks PrimPol[17]. In mammalian cells, with Pol α constantly re-priming on the lagging strand and possibly also the leading strand, one would expect at most half of gap- and RAD51 foci formation to be PrimPol-dependent. However, PrimPol depletion almost completely abrogates the BPDE-induced RAD51 foci formation, recombination, and SCE formation, suggesting that PrimPol is required for far more than 50% of lesion-induced HR events. We can only speculate that either Pol α maybe somehow inhibited at bulky lesions in mammalian cells, or that PrimPol might have additional roles in promoting HR at gaps, for example by interacting with DNA repair factors or due to its preference to synthesize DNA primers[12]. Interestingly, recent work also suggests that different DNA damage tolerance pathways may be active at the leading versus the lagging strand in budding yeast[44]. It will be important, if challenging, to further investigate how re-priming via PrimPol specifically promotes HR-dependent gap repair.

While we have shown that PrimPol re-priming initiates HR at UV, 4-NQO, and BPDE DNA adducts, it will be interesting to investigate whether other bulky lesions or replication impediments can be channeled into this pathway. DNA-protein cross-links, for example, engage HR for repair[45,46] and lead to template-switching[47] raising the question of whether PrimPol could act at these lesions as well. PrimPol also re-primes at R-loops and secondary structure-forming sequences[48,49], opening up the possibility that HR in those backgrounds might depend at least partially on PrimPol. It will be important to test how commonly PrimPol promotes recombination at other replication-blocking structures.

Understanding the molecular mechanism of HR induced by bulky adducts is important for human health. We suggest that this will be particularly useful to interpret more recent cancer genomics data. Both BPDE-treated cells and lung cancers harbor insertion and deletion mutations (indels) of unknown origin. These are in addition to the point mutations ascribed to error-prone TLS[20,50]. Indels could result from replication-associated HR, such as at restarting forks[51], or from defective HR, such as in BRCA-mutant cancers[52]. It is therefore important to decipher which HR pathways are actually induced by bulky adducts. Furthermore, genetic variants in the HR genes *BRCA2* and *RAD52* have been liked to lung cancer susceptibility[53,54]. It will be important to investigate the impact of such genetic variants on HR at ssDNA gaps. A *PRIMPOL* variant has also been suggested to play a potential role in cancer[55].

Taken together, our data support that a large fraction of mammalian replication-associated HR can occur after re-priming, post-replicatively behind the fork.

## Methods

**Cell lines and reagents.** U2OS cells were used as the default cell line unless indicated otherwise. U2OS, A549, and MRC5 cells were obtained from ATCC. The SW480SN.3[56], PrimPol[−/−] MRC5[40], and U2OS cell lines carrying doxycycline (Dox)-inducible PrimPol shRNA[11] have been described before. U2OS, U2OS shPrimPol, A549, and MRC5 were authenticated using 17-locus STR profiling. Cells were confirmed to be free of Mycoplasma infection and grown in Dulbecco's modified Eagle's Medium (DMEM) with 10% fetal bovine serum in a humidified atmosphere containing 5% $CO_2$. MRC WT and PrimPol[−/−] cells were grown in Minimum Essential Media (MEM) with 15% fetal bovine serum. The media was additionally supplemented with L-Glutamine (2 mM), Penicillin (50 U/mL), and Streptomycin (50 μg/mL) for all cell lines. Additionally, SW480SN.3 cells were grown in hygromycin (50 μM) to maintain the SCneo recombination reporter. Mirin, PFM01, 4-nitroquinolie 1-oxide (4-NQO), etoposide, and hydroxyurea (HU) were from Sigma-Aldrich.

**BPDE treatment.** (+)-*anti*-B[a]P-7R,8S-dihydrodiol-9S,10R-epoxide (enantiopure, BPDE) was custom-synthesized by the Biochemical Institute for Environmental Carcinogens (Grosshansdorf, Germany). For long-term storage, BPDE was solved in a mixture of 95% tetrahydrofuran (anhydrous, inhibitor-free) and 5% tri-methylamine (subsequently termed "solvent") to a concentration of 1.65 mM. Aliquots were stored in amber glass vials at −80 °C. Freshly thawed and diluted BPDE aliquots were used to avoid inactivation of the reactive compound BPDE. Following BPDE treatment, media was removed, and cells were washed with PBS twice before culturing in fresh media. Solvent treatment was for the same length as BPDE treatment, and release from solvent was for 0 h unless indicated otherwise.

**Immunofluorescence.** Cells were fixed with 4% PFA for 10 min and permeabilized with 0.25% triton X-100 for 5 min at 4 °C followed by blocking with 4% FCS in PBS. Primary antibodies were rabbit polyclonal anti-RAD51 (Abcam ab63801, 1:800 or Calbiochem PC130, 1:2,000 for IR-induced RAD51 foci), mouse mono-clonal anti-phospho-Histone H2AX (Ser139) (JBW301, Merck 05-636, 1:1,000), rabbit polyclonal anti-53BP1 (Bethyl A300-272A, 1:15,000), mouse anti-RPA32 (Merck NA18, 1:500), rat anti-BrdU (BU1/75, Abcam ab6326, 1:250) to detect CldU. Secondary antibodies were anti-rabbit IgG AlexaFluor 594 (ThermoFisher, 1:500) and anti-mouse IgG AlexaFluor 488 (ThermoFisher, 1:500). For RPA staining, cells were pre-extracted with CSK buffers (CSK1: 10 mM PIPES, 300 mM sucrose, 100 mM NaCl, 3 mM $MgCl_2$, CSK2: 0.5% Triton X-100) on ice previous to PFA fixation. For RAD51 foci detection in response to IR irradiation, cells were also pre-extracted and subjected to EdU staining to label replicating cells. To this end, cells were incubated with EdU at a final concentration of 1 μM for 30 min before staining was carried out as detailed in Click-iT EdU Imaging Kits (Life Technologies). For co-localization staining, primary and secondary antibodies against phospho-Histone H2AX were fixed for 10 min with 2% PFA before DNA denaturation with 2 M HCl for 40 min and immunostaining for CldU. DNA was counterstained with DAPI. Based on the background numbers of foci observed in unchallenged U2OS cells, cells with more than 5 RAD51 foci (10 foci for IR-induced RAD51 staining) and 8 γH2AX or 10 53BP1 foci were scored as positive. Foci were quantified by eye directly on the microscope, and representative images were taken for illustration.

**DNA fiber analysis.** U2OS cells were pulse-labeled with 25 μM CldU and 250 μM IdU, and treated with BPDE for the times indicated. Labeled cells were harvested and

DNA fibers were spread by mixing 2 μl cells with 7 μl spreading buffer (200 mM Tris–HCl pH 7.4, 50 mM EDTA, 0.5% SDS). DNA fiber spreads were fixed in 3:1 methanol:acetic acid. Rehydrated spreads were denatured in 2.5 M HCl for 80 min, incubated with rat anti-BrdU (BU1/75, Abcam ab6326, 1:700) and mouse anti-BrdU (B44, Becton Dickinson 347580, 1:500) for 1 h, fixed with 4% PFA and incubated with anti-rat AlexaFluor 555 (ThermoFisher, 1:500) and anti-mouse AlexaFluor 488 (ThermoFisher, 1:500) for 1.5 h. Fibers were examined using a Nikon E600 microscope with a Nikon Plan Apo 60x (1.3 NA) oil lens, a Hamamatsu digital camera (C4742-95), and the Volocity acquisition software (Perkin Elmer). For the quantification of replication structures, at least 190 structures were counted per experiment. Stalled forks were defined as CldU tracts not followed by an IdU tract. For quantification of fork speeds, the lengths of labeled tracks were measured using the ImageJ software (http://rsbweb.nih.gov/ij/)[57] and arbitrary-length values were converted into μm using a micrometer slide for calibration. Fork stalling was quantified either by scoring percentages of CldU-only fibers of all red-labeled structures or by comparing the lengths of IdU tracks of bidirectional forks (fork asymmetry).

**S1 endonuclease (S1)-modified DNA fiber assay.** U2OS cells were pulse-labeled with 25 μM CldU and 250 μM IdU, and treated with BPDE for the times indicated followed by permeabilization with CSK100 buffer (100 mM NaCl, 10 mM MOPS pH 7.0, 3 mM MgCl$_2$, 300 mM sucrose, 0.5% triton X-100) for 10 min. Nuclei were subsequently treated with either 20 U/mL S1 nuclease (Invitrogen, 18001016) to induce DSBs at sites of DNA gaps or mock-treated (S1 buffer: 30 mM sodium acetate, 10 mM zinc acetate, 5% glycerol, 50 mM NaCl, pH 4.6) for 30 min at 37 °C. Nuclei were harvested by scraping and DNA fiber spreads were prepared, stained, and analyzed as described for the unmodified fibers assay above. At least 120 fibers per condition were measures from 3 independent biological repeats.

**Single-molecule analysis of resection tracks (SMART).** U2OS cells were treated with 20 μM BrdU for 48 h. After treatment with BPDE, siRNA, etoposide, or MRE11 inhibitor as applicable, DNA was spread as for the unmodified DNA fiber assay. Native fiber spreads were stained with mouse anti-BrdU (B44, Becton Dickinson 347580, 1:500) for 1 h, fixed with 4% PFA, and incubated with anti-mouse AlexaFluor 488 (Thermo Fisher, 1:500) for 1.5 h. Fibers were examined as described above. For quantification of DNA gap sizes, the lengths of green (AF 488) labeled native patches were measured using ImageJ and arbitrary-length values were converted into μm using the scale bars created by the microscope. A total of at least 550 fiber stretches derived from 3 independent biological repeats were measured.

**Pulsed-field gel electrophoresis (PFGE).** To detect DSBs, $2 \times 10^6$ cells per sample were treated as indicated, harvested, and melted into 1.0% InCert-Agarose (Lonza) inserts. Inserts were digested in 0.5 M EDTA-1% *N*-laurylsarcosyl-proteinase K (1 mg/ml) at room temperature for 48 h and washed three times in TE buffer. Inserts were loaded onto a separation gel (1.0% chromosomal-grade agarose, Bio-Rad). The separation was performed using a CHEF DR III (BioRad; 120 field angle, 240 s switch time, 4 V cm$^{-1}$, 14 °C) for 20 h. Images of ethidium bromide-stained gels were acquired using a Syngene G:BOX gel imaging system. DSBs (chromosome fragments >2 Mbp) were quantified by densitometry using ImageJ and normalized to the total amount of DNA in the gel.

**siRNA and DNA transfection.** Custom made siRNAs against RAD51[58], and PrimPol[12] were from Dharmacon, and custom made EXO1 siRNA (Sense: GAA CAA GGU UCC UGG GCU AUA[dT][dT], Antisense: [Phos]UAU AGC CCA GGA ACC UUG UUC[dT][dT]) was from Sigma-Aldrich. "Allstars negative control siRNA" was from Qiagen. Cells were transfected for 48 h with 50 nM siRNA using Dharmafect 1 reagent (Dharmacon). Expression of PrimPol shRNA was induced with 1 μg/ml Dox for 3 d. For rescue experiments, 2.5 μg of PrimPol variant expressing pcDNA3.1/nV5-DEST vectors[11] or control plasmid pEGFP-C2 (Clontech) were transfected for the last 24 h using TransIT-2020 (Mirus Bio). To generate siRNA-resistant expression vectors, pcDNA3.1/nV5-DEST vectors expressing PrimPol WT or CH variant[11] were subjected to site-directed mutagenesis using forward primer CGT CTG TGT ACA GAC CAA GAT TGT CCA AG; reverse primer ACA AGG GCT TTC TCT CAT AAT GAG ATG CTC and the Q5® Site-Directed Mutagenesis Kit (New England Biolabs). Cells were transfected with 50 nM siRNA using Dharmafect 1 for 48 h, and with 2.5 μg siRNA-resistant expression vectors or pEGFP-C2 using TransIT-2020 for the last 24 h.

**Western blotting.** Cell extracts were prepared in UTB buffer (50 mM Tris–HCl pH 7.5, 150 mM β-mercaptoethanol, 8 M urea) and sonicated to release DNA-bound proteins. Primary antibodies used were rabbit anti-RAD51 (Abcam, ab63801, 1:1,000), rabbit anti-EXO1 (Bethyl, A302-640, 1:1,000), rabbit anti-PRIMPOL (described in ref. [11], 1:1000), rabbit anti-SMC1 (a kind gift from A. Losada's lab, CNIO[59], 1 μg/ml), mouse anti-V5 tag (Invitrogen, R960-25, 1:1:500), and mouse anti-αTUBULIN (B512, Sigma T6074, 1:10,000).

**Quantitative RT-PCR.** Total RNA was harvested using the miRNeasy Mini Kit (Qiagen) followed by DNase I treatment (Roche). 1 μg of total RNA was reverse-transcribed using SuperScript Reverse Transcriptase III (Thermo Fisher) with random primers (Promega), following manufacturer's instructions. The qPCR primers for amplification are listed in Supplementary Table 1. For quantitative RT-PCR, 2 μl of cDNA were analyzed using a CFX Connect real-time PCR machine (BioRad) with SensiFAST SYBR Lo-ROX Kit (Bioline). Cycling parameters were 95 °C for 3 min, followed by 40 cycles of 95 °C for 10 s, 60 °C for 30 s. The result was normalized to RPLP0. ΔcT was calculated as the difference in the cycle threshold of the transcript of interest and RPLP0, plotted as fold change compared to the CTR untreated sample.

**Recombination in SW480SN.3 cells.** Following siRNA treatments, SW480SN.3 cells were rinsed in PBS and left in fresh non-selective media for 48 h. Cells were treated with solvent or 50 nM BPDE for 20 min, rinsed in 2x PBS, trypsinized and directly reseeded. To determine cloning efficiency, two dishes were plated with 500 cells each. For the selection of recombinants cells were grown in G418 (1 mg/ml; 25 cells/mm$_2$) in two technical repeats. After 14–18 days the colonies obtained were stained with methylene blue in methanol (4 g/l) and the number of recombinants was calculated per $10^{-5}$ colony-forming cells. Recombination frequencies were normalized to BPDE-treated samples to account for the high variation in spontaneous recombination between samples.

**Analysis of sister chromatid exchange (SCE).** U2OS were incubated with 240 μM BrdU for two cell cycles and treated with BPDE, 4-NQO, and siRNA as indicated. 4 h before trypsinization, 0.02 μg/mL colcemid (Sigma-Aldrich) was added. In total 0.075 M KCl hypotonic buffer was added to pelleted cells for 15 min and cells were then fixed in 3:1 methanol:acetic acid. Spreads were prepared by dropwise addition to a glass slide and left to dry in the dark overnight. Slides with metaphases were stained in 10 μg/mL Hoechst 33258 in water for 20 min, exposed to long-wave (365 nm) UV light in a glass dish of 2x SSC buffer (30 mM sodium citrate, 300 mM NaCl in water) for 8–10 min, and stained in 20% Leishman's stain (VWR) diluted in Gurr buffer (Gibco) for 6 min at room temperature. Slides were analyzed using a Zeiss Axio Observer microscope. SCEs were defined by an exchange of dark-stained (Hoechst-bound TT-rich) segments with light, bleached (BrdU incorporated) segments. Staining variation where chromatids clearly twisted around each other was excluded from the count. For each condition, at least 90 metaphases from at least 3 independent biological repeats were analyzed. SCE rates were calculated per diploid set of chromosomes to compensate although U2OS present with chromosome counts in the hypertriploid range. SCEs were quantified directly on the microscope, and representative images were taken for illustration.

**Colony survival assay.** U2OS cells were siRNA transfected for 24 h previous to being plated in triplicates for both 500 and 1000 cells. Cells were given 24 h to adhere before treatment with BPDE (10–400 nM) for 20 min. Subsequently, cells were washed 2x PBS and colonies of >50 cells were allowed to form in fresh medium and fixed in 50% ethanol, 2% methylene blue.

**Flow cytometry.** U2OS cells were transfected with siRNA for 48 h followed by treatment with 50 nM BPDE and release into fresh DMEM. Cells were incubated with 100 μM BrdU for the last 30 min of the indicated release times and then fixed in 70% ethanol overnight. Cells were incubated in 2 M HCl/0.1 mg/ml pepsin for 30 min. Samples were washed in PBS before incubation with mouse anti-BrdU (B44, Becton Dickinson 347580, 1:200) for 1 h and subsequently with anti-mouse IgG AlexaFluor 488 (ThermoFisher) for 1 h. Finally, cells were incubated with 10 μg/ml propidium iodide and 25 μg/ml RNaseA for 30 min. Cell cycle profiles were gathered using the BD LSR Fortessa ×20 and analyzed with BD FacsDiva software.

**Quantification and statistical analysis.** Column graphs represent the means + 1x SEM of at least 3 independent biological repeats. Scatter graphs show individual data points from at least 3 independent biological repeats. The number of independent biological repeats (n) is indicated in the figure legends. For foci analysis, at least 10 different areas with at least 140 cells in total were quantified for each independent biological repeat. Statistical tests were performed using Graphpad Prism v6-8 and Microsoft Excel v16. The statistical tests used throughout were the one-sided student's *t*-test, one-sided Mann–Whitney test, or one-way ANOVA followed by Dunnett, Tukey, or Sidak test. Asterisks compare to control, unless indicated otherwise by lines or in the figure panels, and signify *$p < 0.05$, **$p < 0.01$, ***$p < 0.001$. ****$P < 0.0001$.

**Reporting Summary.** Further information on research design is available in the Nature Research Reporting Summary linked to this article.

## Data availability

Raw data for this study mainly comprise microscopy images of the DNA fiber assay. These images and all relevant data are available from the authors upon reasonable request. A reporting summary for this article is available as a Supplementary Information file. Source data are provided as a Source Data file. Source data are provided with this paper.

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

## Acknowledgments

We thank Drs Rebecca M. Jones, Katarzyna Starowicz, and Karen Sisley for advice on PFGE and SCE quantification, and Dr. Ana Losada for the kind gift of anti-SMC1 antibody. This work was supported by the German Research Foundation Pl 1300/1-1 (A.L.P.), Medical Research Council MR/S021310/1 (A.L.P. and E.P.), Cancer Research UK C25526/A28275 and Cancer Research UK Birmingham Centre Award C17422/A25154 (E.P.), Cancer Research UK C8820/A19062 (A.K.W. and J.R.M.), Wellcome Trust 206343/Z/17/Z (J.R.M.), University of Sheffield 322149 (H.E.B.), Biotechnology and Biological Science Research Council BB/H019723/1 and BB/M008800/1 (A.J.D.), and the Spanish Ministry of Science and Innovation BFU2016-80402-R, co-financed by E.U. ERDF funds (J.M.).

## Author contributions

A.L.P. and E.P. conceived the study; A.L.P., A.B., R.D.W.K., A.K.W., D.G.-A., L.J.B., and H.E.B. performed the experiments; A.L.P., A.K.W., J.M., A.J.D, J.R.M., H.E.B., and E.P. designed the experiments; A.L.P. and E.P. wrote the paper.

## Competing interests

The authors declare no competing interests.
