## [Peer Review File · Nature Communications]

Reviewers' comments:

Reviewer #1 (Remarks to the Author):

The manuscript "PrimPol-dependent single-stranded gap formation mediates homologous recombination at bulky DNA adducts" by Piberger et al. establishes the role of RAD51-mediated homologous recombination (HR) in post-replicative repair of bulky DNA adducts. The authors demonstrate that exposure of human U2OS cells to low concentrations of BPDE, an active metabolite of cigarette smoke carcinogen benzo[a]pyrene (BP) that forms bulky DNA adducts with predominantly N2 amino group of guanine, does not stall DNA replication. Significantly, they observed recruitment of RAD51 to single stranded (ss) DNA gaps induced by the BPDE treatment in the absence of formation of double strand breaks (DSB). This is in contrast to the exposures to high BPDE concentrations that resulted in DSBs accompanied by RAD51 foci. Further, the authors employed fibre assay to show that the ssDNA gaps produced under low BPDE concentration were generated behind the progressing replication fork. Moreover, the authors conducted experiments to demonstrate that the primase activity of PrimPol is required for RAD51 loading under the low BPDE concentration. The authors thus reached the conclusion that HR can occur as a response to the low numbers of BPDE-DNA adducts in absence of stalled or collapsed replication forks at ss gaps produced by the primase activity of PrimPol.

Overall, the functional roles of RAD51-mediated HR activation in response to low concentration of BPDE exposure and the necessity of PrimPol in re-priming in this process are shown for the first time. These findings are very important for understanding of the interplay of lesion tolerance and repair mechanisms in human cells. The experiments are well performed and described and the manuscript is clearly written. I would strongly recommend publishing the article in Nature Communications with minor corrections as follows:

1. Line 91. "we used BPDE lesions.." should be "BPDE-DNA lesions or BPDE-induced DNA lesions".
2. The solvent for BPDE solution preparation is not described in the Methods section. Please, clarify how BPDE solution was made and how the cells were exposed to it and washed out from it.

Reviewer #2 (Remarks to the Author):

In this manuscript, Piberger et al. report an analysis of PrimPol-mediated re-priming and its consequences after bulky DNA lesions. They observe that non-lethal concentrations of the adduct-forming BPDE do not cause replication fork stalling, but rather the formation of postreplicative daughter-strand gaps, and that gap formation is promoted by PrimPol activity. They show that efficient loading of the recombination factor RAD51 requires processing of these structures by EXO1 and MRE11. Although RAD51 is needed for their resolution, it is not required for the re-priming or gap formation itself. The authors then expand their analysis to other bulky lesions caused by UV-C or 4-NQO and show that a similar principle applies. They also differentiate the mechanism from the processing of double-strand breaks, which do not require PrimPol for RAD51 loading, and they show that PrimPol depletion also reduces recombination and sister chromatid exchange, suggesting that such events emerge from daughter-strand gaps rather than stalled replication forks.

The findings reported in this manuscript are exciting and novel. Whereas daughter-strand gap formation is well accepted in budding yeast and more and more publications highlight the importance of such structures for checkpoint activation and damage processing, their relevance has remained unclear in mammalian cells, where textbook models generally assume that damage processing happens at replication forks. This manuscript refutes that model for the case of bulky adducts, thus emphasizing the relevance of events that occur in a manner uncoupled from the replication fork. Therefore, the study makes a very important contribution to the field that will

change the view on how replication stress is caused by DNA damage.

There are a number of technical issues and discussion points that would be worth addressing in a revised manuscript:

Major points:

1. One salient question that the authors only cursorily discuss is the relevance of PrimPol for lesions on the lagging strand. As they correctly state in the Discussion, they would expect that PrimPol is needed for at most 50% of all lesions. Is there any way the absolute requirement can be extracted from their data? At present, they - correctly - describe that PrimPol "promotes" gap formation and RAD51 loading etc., but it does not become clear to what extent this process contributes when considering the total number of lesions. I understand that this might be difficult to assess in light of alternative pathways such as fork regression or TLS, but it would be important to consider this point in more detail.

2. Quantification of foci needs to be described more accurately. The number of foci used for evaluation (5, 8, 10) appears arbitrary and was not justified; it is unclear whether they applied a threshold to determine what counts as a focus; or did they do that "by eye", and if so, were the samples blinded or not? Were foci counted on a single focal plane or throughout the nucleus? Direct quantification at the microscope seems outdated and possibly subjective nowadays where image analysis programmes and routines are available. At least a detailed description would be required.

3. The S1 nuclease assays (Figs 2C, G) are key to the message of the study. Therefore, they should be better controlled. In particular, the authors should make sure that they can exclude an impact of NER (or BER) on gap formation in the labelled DNA, as these events would result in a shortening of CldU tracts and would skew the C/I ratio. A parallel analysis of the absolute lengths of CldU and IdU tracts would be helpful here, and the ratio should also be examined under condition of NER or BER inhibition.

Measuring absolute tract lengths should also give information about fork slowing (of non-stalled forks) in the presence of BPDE.

Finally, what is the propensity of gaps to appear within the CldU tract. These gaps would give rise to truncated CldU tracts (which would probably not be counted in this analysis - but how frequent are they compared to dual-labelled tracts?) as well as tracts that start with a bit of BrdU and continue with an IdU label (i.e. the "other end"). How will these structures be accounted for? Do the authors differentiate between labels at what end of the fibre the label sits (IdU at the tip or CldU at the tip)?

4. The situation in response to 500 nM BPDE seems unresolved. At this dose, survival is close to zero and 53BP1 foci accumulate; yet, the analysis in Fig. S1E does not reveal any significant DSB formation. How do cells die under these conditions? The authors postulate that a damage response might be inhibited at high BPDE. But gH2AX signalling is observable. Please comment on this.

Minor points:

5. Figure legends should generally be more explicit and provide a bit more explanation. Examples:

- Figs 1C, E (and others): at what time was the "solvent" sample taken?
- Figs 1F, 2F (and others): explain that stalled forks are defined as the percentage of CldU tracts not followed by an IdU tract (I assume that this is how the values were generated).
- Fig. S1B: the axis would better be labelled as "Average ..." if the bar represents the mean.
- In Fig. 5F, the "%" label is missing on the Y axis.
- Labelling of controls appears confusing sometimes, due to the use of "solvent", "mock", "DMSO", or "0" - please make sure these labels are consistent and clearly defined.
- Please make sure to mention which cell type was used for each experiment (or include a

statement about the default cell line used unless otherwise indicated).

6. Line 68: It is not entirely appropriate to state that damage tolerance can occur via re-priming or TLS. These two options are not alternatives, because any re-priming would merely delay the problem of damage tolerance, and a damage bypass pathway (template switching, HR or TLS) would still need to occur after re-priming to fill the gap.

7. Line 195: shouldn't this be "skewed downwards"? And compared to what?

8. Figs 2C and S3D: why is there a drop in the C/I ratio upon S1 treatment in the absence of siRNA or BPDE? Does that mean spontaneous ssDNA gaps are more prevalent in the CIdU tracts?

9. Figs 3B and 1D appear redundant. Could they not be combined?

10. Fig 4: One would expect that EXO1 and MRE11 activities are additive, but that their contributions depend on PrimPol. Could this be tested? As a control, the authors should also show how long the native BrdU tracts are under PrimPol or EXO1 depletion in undamaged conditions.

11. Fig. 2A: Analysis of RPA32 foci shows a rapid increase after release from BPDE. As in the previous experiments, it would be good to show the situation immediately after the BPDE treatment (i.e. without time for recovery). The same applies to Figs 5C, 5D, 5E, 5F, S2A, S3B, S4A/B/F unless the authors show in a representative case that no increase is observable at this early time).

12. Would it be instructive to use a TLS-deficient PrimPol mutant (compare Kobayashi et al, Cell Cycle 2016) to differentiate a possible contribution of the enzyme's TLS activity from its re-priming function?

13. Fig. 1E: A cell cycle analysis under high and low BPDE conditions would be useful to examine whether a G1 arrest at high BPDE would explain why the number of RAD51 foci doesn't increase under those conditions.

14. Fig. 6F: The model neglects potential TLS events at daughter-strand gaps as well as recombination events that do not result in sister chromatid exchanges. Also, the arrow indicating strand invasion seems to imply that a terminus needs to be unwound adjacent to the gap. It is more likely that the RAD51-covered ssDNA itself would invade the duplex directly, without involving an end.

15. Line 390: One thought on the ability of PrimPol to use dNTPs for priming: given the large excess of rNTPs over dNTPs in cells, how likely is it that the enzyme makes DNA primers in physiological conditions?

Reviewer #3 (Remarks to the Author):

This manuscript examines the replication-associated responses to bulky lesions caused by BPDE. The authors conclude that these lesions are processed during replication via post-replicative gap repair. The gaps are generated by PolPrim-dependent repriming and enlargement by exonuclease activities. This leads to RAD51 loading which presumably is involved in the gap repair mechanism. Overall, I found the manuscript to be interesting and the experiments to be performed to high standards. The general concept being advanced (that DNA damage is often bypassed by forks via new primer synthesis leaving post-replicative gaps that can then be repaired) is not new, nor is the involvement of PrimPol in these types of pathways new, but the manuscript does provide more mechanistic information and the first extensive look at these mechanisms in response to the specific BPDE lesions in human cells. In some cases the authors conclusions are overstated. For

example, in the abstract they write “bulky adducts predominantly induce HR...” I don’t think the data is consistent with the word “predominantly”. The relative contributions of the described pathway vs. TLS vs fork reversal as each fork encounters a BPDE lesion are not really addressed. The statement that RAD51 is not recruited to stalled forks may be incorrect. The authors are using a measurement of RAD51 localization (foci formation), that requires a specific threshold of molecules be recruited to a cluster of replication forks within a factory to be visualized. It is reasonable to expect that more sensitive assays could reveal localization of RAD51 even to stalled forks when foci are not readily discernable. For example, PLA analyses have been used by other investigators to detect RAD51 localization to stalled forks.

Other comments:

1. The authors should tone down some of their conclusions especially when they are arguing based on negative results.
2. The increase in stalled forks in response to PrimPol depletion is actually very small (from about 8% to 16% in Fig 2F). Doesn’t this indicate that most (84% at least) of the forks do not stall even when PrimPol is depleted and/or other pathways like TLS could predominate?
3. What would be the expected density of lesions caused by BPDE at this concentration? Do most forks never encounter a lesion? Would they generally only encounter a lesion on one of the template strands? A measurement of the actual lesion density if it is not known from the literature is needed to understand the results.
4. The Vindigni lab reported that PrimPol function and expression are time-dependent in response to cisplatin lesions (Quinet 2020). Is this true for 4NQO?
5. The continued RPA32 foci in RAD51 depleted cells after release from BPDE (Fig 2A) is interpreted as an involvement of RAD51 in resolution of the ssDNA. How do we know this is not because RAD51 is not available to replace RPA?
6. More generally, the manuscript would be strengthened by additional data on resolution of damage and ssDNA gaps that formed. This is important since the authors are arguing that these gaps are eventually repaired via a post-replicative mechanism.
7. It is striking that the authors do not examine markers of template switching or the requirements for these mechanisms in their analyses like PCNA ubiquitylation. Those analyses would strengthen the manuscript by adding additional mechanistic understanding of the pathway.

Response to reviewers' comments (Piberger et al.)

We thank the reviewers for their positive assessments and useful suggestions for improving our manuscript. We have made the requested textual changes and have added 9 new figure panels to the supplemental material.

Under normal circumstances, we would have performed more additional experiments. But because our lab was closed down due to coronavirus, and the first author is still unable to return to the bench, we have selected some key experiments that could be realistically completed under these conditions. Below we explain how we have addressed the reviewers' queries amid this reduced ability to perform additional lab work.

Reviewer #1:

1. Line 91. "we used BPDE lesions.." should be "BPDE-DNA lesions or BPDE-induced DNA lesions".

We have changed this to "BPDE-DNA lesions" (line 89).

2. The solvent for BPDE solution preparation is not described in the Methods section. Please, clarify how BPDE solution was made and how the cells were exposed to it and washed out from it.

We are now describing BPDE solution preparation in the Methods (line 471): "For long-term storage, BPDE was solved in a mixture of 95% tetrahydrofuran (anhydrous, inhibitor-free) and 5% trimethylamine (subsequently termed 'solvent') to a concentration of 1.65 mM. Aliquots were stored in amber glass vials at -80°C. Freshly thawed and diluted BPDE aliquots were used to avoid inactivation of the reactive compound BPDE. Following BPDE treatment, media was removed, and cells were washed with PBS twice before culturing in fresh media. Solvent treatment was for the same length as BPDE treatment, and release from solvent was for 0 h unless indicated otherwise."

Reviewer #2:

Major points:

1. One salient question that the authors only cursorily discuss is the relevance of PrimPol for lesions on the lagging strand. As they correctly state in the Discussion, they would expect that PrimPol is needed for at most 50% of all lesions. Is there any way the absolute requirement can be extracted from their data? At present, they - correctly - describe that PrimPol "promotes" gap formation and RAD51 loading etc., but it does not become clear to what extent this process contributes when considering the total number of lesions. I understand that this might be difficult to assess in light of alternative pathways such as fork regression or TLS, but it would be important to consider this point in more detail.

We can only estimate the relevance of PrimPol at lesions that activate HR, which is less than the total number of lesions. For illustration, 50 nM BPDE should induce 10,000 lesions per genome and we only observe 60-80 RAD51 foci per cell at any point (Figure S1D). It is also known that some BPDE lesions are repaired by NER or may indeed be bypassed by TLS. However, PrimPol depletion almost completely abrogates the BPDE-induced increases in HR readouts i.e. RAD51 foci, recombination reporter and SCEs, suggesting that PrimPol is required for far more than 50% of lesion-induced HR events. We do not understand this at present but speculate that either DNA Pol alpha is for some reason not able to reprime at bulky lesions in mammalian cells or that PrimPol might have additional roles in promoting HR at gaps, for example by interacting with important pathway choice factors.

We have now added these additional considerations the Discussion (line 412): "In mammalian cells, with Pol α constantly re-priming on the lagging strand and possibly also the leading strand, one would expect at most half of gap- and RAD51 foci formation to be PrimPol-dependent. However, PrimPol depletion almost completely abrogates the BPDE-induced RAD51 foci formation, recombination and SCE formation, suggesting that PrimPol is required for far more than 50% of

lesion-induced HR events. We can only speculate that either Pol α is somehow inhibited at bulky lesions in mammalian cells, or that PrimPol might have additional roles in promoting HR at gaps, for example by interacting with DNA repair factors or due to its preference to synthesise DNA primers. Interestingly, recent work also suggests that different DNA damage tolerance pathways may be active at the leading versus the lagging strand in budding yeast⁴². It will be interesting, if challenging, to further investigate how re-priming via PrimPol specifically promotes HR-dependent gap repair.“

2. Quantification of foci needs to be described more accurately. The number of foci used for evaluation (5, 8, 10) appears arbitrary and was not justified; it is unclear whether they applied a threshold to determine what counts as a focus; or did they do that "by eye", and if so, were the samples blinded or not? Were foci counted on a single focal plane or throughout the nucleus? Direct quantification at the microscope seems outdated and possibly subjective nowadays where image analysis programmes and routines are available. At least a detailed description would be required.

The foci were counted by eye and throughout the nucleus. We find this to be more sensitive and the foci clearer to detect by eye than in images, especially RAD51 foci because they are smaller. The thresholds for counting foci-positive cells were defined similarly to what would be done using image analysis software. Based on the background numbers of foci that can be routinely observed in unchallenged U2OS cells; we find that they rarely have more than 8-9 γ H2AX and 53BP1 foci, or more than 4 RAD51 foci. Increases in numbers after DNA damaging treatments then show where foci have been induced by the treatment. Investigators were not blinded to group allocation during data collection and analysis. This was because we thought that blinding was not necessary to avoid bias, mainly because we are looking at fairly large effects and because results have been reproduced independently by different investigators in the group, including with using blinding. This information is in the Reporting Summary.

We have also added a description in Methods (line 499): “Based on the background numbers of foci observed in unchallenged U2OS cells, cells with more than 5 RAD51 foci (10 foci for IR-induced RAD51 staining) and 8 γ H2AX or 10 53BP1 foci were scored as positive. Foci were quantified by eye directly on the microscope, and representative images were taken for illustration.”

3. The S1 nuclease assays (Figs 2C, G) are key to the message of the study. Therefore, they should be better controlled. In particular, the authors should make sure that they can exclude an impact of NER (or BER) on gap formation in the labelled DNA, as these events would result in a shortening of CldU tracts and would skew the C/I ratio. A parallel analysis of the absolute lengths of CldU and IdU tracts would be helpful here, and the ratio should also be examined under condition of NER or BER inhibition. Measuring absolute tract lengths should also give information about fork slowing (of non-stalled forks) in the presence of BPDE.

We agree that the S1 nuclease assays are important for the message, although we would argue that they are also complemented by SMART assay data.

We have now added the absolute lengths of CldU and IdU tracts for all S1 nuclease assays in Supplementary Figures 3A, B, E, F and 6A, B. These do not suggest that there are gaps particularly in the CldU tracks after BPDE treatment (Fig. S3B, F, see also response further below). They also suggest that there is only minor slowing of non-stalled forks in the presence of BPDE, which is also supported by the data in Fig. S1E.

If excision repair pathways contribute to S1 fibre shortening, this should occur in both labels and with and without PrimPol, thus not affecting the conclusions. It seems unlikely that PrimPol performs an unknown role in NER or BER, as there is no need for repriming in excision repair as the 5' end of the gap or nick serves as the primer, and other DNA polymerases fill the gaps. We have added the following to the Results section (line 227): “One caveat is that we did not perform the S1 nuclease fibre assay also under conditions of inhibiting NER and base excision repair, which both could induce ssDNA gaps and contribute to S1 fibre shortening after treatment with BPDE. However,

excision repair-induced gaps would occur independently of PrimPol, thus not affecting the overall impact of PrimPol depletion.”

Finally, what is the propensity of gaps to appear within the CldU tract. These gaps would give rise to truncated CldU tracts (which would probably not be counted in this analysis - but how frequent are they compared to dual-labelled tracts?) as well as tracts that start with a bit of BrdU and continue with an IdU label (i.e. the "other end"). How will these structures be accounted for? Do the authors differentiate between labels at what end of the fibre the label sits (IdU at the tip or CldU at the tip)?

We do not detect an induction of on gaps in the CldU tracts after BPDE treatment, probably due to low lesion density. The likelihood of forks encountering damage is higher during the IdU pulse labelling period, because it is 2.5 times longer. We did quantify truncated CldU tracts after treatment with BPDE and S1 nuclease but found no increase in these structures (see graph below). We observe no consistent S1-induced shortening of CldU tracts across experiments, suggesting there are few structures with cut CldU tracts (Fig. S3A, B, E, F).

Figure: Percentage of CldU-only tracts in U2OS cells after treatment with 50 nM BPDE +/- S1 nuclease as in Figure 2C. N=2. Bars: means +/- standard deviation.

4. The situation in response to 500 nM BPDE seems unresolved. At this dose, survival is close to zero and 53BP1 foci accumulate; yet, the analysis in Fig. S1E does not reveal any significant DSB formation. How do cells die under these conditions?

It seems clear that the cells are dying of DNA damage overload after 500 nM BPDE, through apoptosis or mitotic catastrophe (see e.g. PMID: 27694624). The lack of DSBs in the PFGE analysis does not rule out DSB formation because that analysis was performed only 3 hours after release from BPDE, and DSBs may accumulate later. Our 53BP1 foci analysis in Figure 1G suggests that DSBs accumulate from 6 hours after release from 500 nM BPDE.

The authors postulate that a damage response might be inhibited at high BPDE. But γ H2AX signalling is observable. Please comment on this.

It is possible for the ATM-mediated DSB response downstream of γ H2AX to be impaired, even when γ H2AX itself is still normally formed (see e.g. PMID:29180822; PMID:30560944). We have aimed to clarify this more by changing the text (Results line 163): “Interestingly, highly cytotoxic treatment with 1.65 μ M BPDE may suppress the DSB response downstream of γ H2AX”.

Minor points:

*5. Figure legends should generally be more explicit and provide a bit more explanation. Examples:
- Figs 1C, E (and others): at what time was the "solvent" sample taken?*

We have added to the Methods section (line 477): “Solvent treatment was for the same length as BPDE treatment, and release from solvent was for 0 h unless indicated otherwise”, and indicated in the figure legends where solvent samples were taken differently.

- Figs 1F, 2F (and others): explain that stalled forks are defined as the percentage of CldU tracts not followed by an IdU tract (I assume that this is how the values were generated).

We have clarified this in the Results (line 146) and Methods sections (line 514).

- Fig. S1B: the axis would better be labelled as "Average ..." if the bar represents the mean.

We have made the requested change.

- In Fig. 5F, the "%" label is missing on the Y axis.

We have made the requested change.

- Labelling of controls appears confusing sometimes, due to the use of "solvent", "mock", "DMSO", or "0" - please make sure these labels are consistent and clearly defined.

We have changed the labelling in Figure 5D and E from to solvent (DMSO) and solvent (water) and clarified that mock refers to mock irradiation in the legends for Figure 5C and F.

- Please make sure to mention which cell type was used for each experiment (or include a statement about the default cell line used unless otherwise indicated).

We have added to the Methods section (line 454): "U2OS cells were used as the default cell line unless indicated otherwise."

6. Line 68: It is not entirely appropriate to state that damage tolerance can occur via re-priming or TLS. These two options are not alternatives, because any re-priming would merely delay the problem of damage tolerance, and a damage bypass pathway (template switching, HR or TLS) would still need to occur after re-priming to fill the gap.

We apologise that this was ambiguously worded. We have changed the wording (line 65) to: "Fork stalling can be avoided through re-priming, for example by PrimPol, a recently described RNA/DNA primase that exerts both DNA/RNA primase and DNA polymerase activity¹⁰ and can re-prime after UV lesions¹¹⁻¹⁴. Damaged DNA can be bypassed either through error-prone translesion synthesis (TLS) that is promoted by PCNA mono-ubiquitination, or through an error free damage tolerance pathway that is promoted by PCNA polyubiquitination and also uses recombination proteins."

7. Line 195: shouldn't this be "skewed downwards"? And compared to what?

The ratios are skewed upwards compared to what would be expected based on the labelling times, because a ratio of 20 min (CldU)/50 min (IdU) would be closer to 0.4 than to 1. We have now modified this sentence (line 202) to: "The IdU pulse is extended to 50 min in order to capture more events, even though this leads to CldU/IdU ratios that are skewed upwards compared to what would be expected based on the labelling times".

8. Figs 2C and S3D: why is there a drop in the C/I ratio upon S1 treatment in the absence of siRNA or BPDE? Does that mean spontaneous ssDNA gaps are more prevalent in the CldU tracts?

The absolute lengths of CldU and IdU tracts for these experiments (Fig. S3A, B, E) show that there is some reduction in CldU lengths, but that IdU lengths are unchanged or increased. We don't believe that there are more spontaneous gaps especially in the CldU tracts, but rather that the assay is not very suitable for detecting spontaneous gaps in the IdU tracts. This could be because the IdU labelling time has to be quite long, leading to faster-moving forks to merge, making them unavailable to measuring. Cutting by S1 at spontaneous gaps could make longer IdU tracts available for measuring again, revealing more faster-moving forks and lowering the CldU/IdU ratio.

9. Figs 3B and 1D appear redundant. Could they not be combined?

We have considered combining these two part-figures (3B and 3D), but because these two datasets were obtained using two different RNAi sequences, which is important to control for RNAi off-target effects, we believe that continuing to show both figures will be more helpful.

10. Fig 4: One would expect that EXO1 and MRE11 activities are additive, but that their contributions depend on PrimPol. Could this be tested? As a control, the authors should also show how long the native BrdU tracts are under PrimPol or EXO1 depletion in undamaged conditions.

This could be tested by combining EXO1 and PrimPol siRNA with MRE11 inhibitors. We would have attempted this without the lockdown, but we think that this experiment is not strictly necessary to support the conclusions. We have added a caveat in the Results (line 282): “MRE11 and EXO1 activities may be additive in promoting the resection of the ssDNA gaps, which we have not tested.”

11. Fig. 2A: Analysis of RPA32 foci shows a rapid increase after release from BPDE. As in the previous experiments, it would be good to show the situation immediately after the BPDE treatment (i.e. without time for recovery). The same applies to Figs 5C, 5D, 5E, 5F, S2A, S3B, S4A/B/F unless the authors show in a representative case that no increase is observable at this early time).

We would argue that adding more very early time points would add more information but is not necessary to support the conclusions.

Firstly, we already show some data that should address this query. Fig. S2E shows that RPA32 foci are only slightly increased 1 hour after 50 nM BPDE and continue to increase between 1 and 2 hours after BPDE. We also already show five datasets with RAD51 foci at 0 hours after 50 nM BPDE, supporting that RAD51 foci are only slightly increased immediately after BPDE treatment (Fig. 1E, 3B, 4G-H, S1D).

Secondly, BPDE is washed out after 20 minutes to keep damage loads low, but there is no specific functional difference between during and after the treatment, as replication forks will continue to encounter unrepaired lesions after release from BPDE and cells do not fully recover until days later. We do not therefore expect the mechanism to be any different after 20 min BPDE compared to 20 min BPDE plus 1 hour release. We would therefore argue that it is not strictly necessary to show 0 hour time points for the qRT-PCR, DNA fibre and FACS analyses.

12. Would it be instructive to use a TLS-deficient PrimPol mutant (compare Kobayashi et al, Cell Cycle 2016) to differentiate a possible contribution of the enzyme's TLS activity from its re-priming function?

This is a great suggestion, but not so likely to produce very clear results. The Y89D mutant cited as TLS defective in Kobayashi et al. still retains its ability to perform TLS, albeit at lower polymerase activity (PMID: 25262353). Furthermore, this mutant seems to act as a dominant negative, slowing unperturbed replication fork progression, which could lead to unwanted skewing of the results.

13. Fig. 1E: A cell cycle analysis under high and low BPDE conditions would be useful to examine whether a G1 arrest at high BPDE would explain why the number of RAD51 foci doesn't increase under those conditions.

This is a very good suggestion and we have now addressed this question by performing BrdU FACS analysis after 500 nM BPDE treatment (new Supplementary Figure 1B, C). This shows that the cells accumulate in S phase rather than G1 after 500 nM BPDE. We have added this to the Results (line 131): “These observations were not simply due to cell cycle changes, as 50 nM BPDE had minimal effects on cell cycle distribution, and 500 nM BPDE induced an S phase arrest rather than a G1 arrest (Supplementary Figure 4A, Supplementary Figure 1B, C)”.

14. Fig. 6F: The model neglects potential TLS events at daughter-strand gaps as well as recombination events that do not result in sister chromatid exchanges.

We have now added a statement about this to the figure legend: “Potential TLS events at gaps and recombination not resulting in sister chromatid exchanges have been omitted for simplicity.” We believe this would be preferable to making a very complex figure.

Also, the arrow indicating strand invasion seems to imply that a terminus needs to be unwound adjacent to the gap. It is more likely that the RAD51-covered ssDNA itself would invade the duplex directly, without involving an end.

We have changed the position of the arrows in Figure 6F and in Figure 2D.

15. Line 390: One thought on the ability of PrimPol to use dNTPs for priming: given the large excess of rNTPs over dNTPs in cells, how likely is it that the enzyme makes DNA primers in physiological conditions?

The excess of NTPs over dNTPs in human cells is between 10-100 fold (PMID: 7877593) but PrimPol displays a large preference for dNTPs over NTPs in presence of Magnesium in vitro (Bianchi et al., Mol Cell 2013) which could compensate for this and still produce DNA primers. We have added the second reference at this point in the discussion.

Reviewer #3:

In some cases the authors conclusions are overstated. For example, in the abstract they write “bulky adducts predominantly induce HR...” I don’t think the data is consistent with the word “predominantly”. The relative contributions of the described pathway vs. TLS vs fork reversal as each fork encounters a BPDE lesion are not really addressed.

We apologise that the emphasis of this sentence was misleading. We have changed it to make clear the intended statement: “we show that HR induced by bulky adducts in mammalian cells predominantly occurs at post-replicative gaps formed by the DNA/RNA primase PrimPol.”

The statement that RAD51 is not recruited to stalled forks may be incorrect (...).

We apologise for the misunderstanding –RAD51 can of course be recruited to stalled forks, as stated in the introduction. However, our data suggest that RAD51 foci formation in response to BPDE is not the result of fork stalling. We have changed the statement in the abstract to: “RAD51 recruitment under these conditions does not result from fork stalling, but rather occurs at gaps formed by PrimPol re-priming and resection by MRE11 and EXO1.”

Other comments:

1. The authors should tone down some of their conclusions especially when they are arguing based on negative results.

We have changed the wording in the section on negative results after RAD51 depletion (lines 167 – 184) and colony forming assays (line 336) to clarify that our data provide no evidence for a number of potential mechanisms, rather than definitely showing that these mechanisms are not relevant.

2. The increase in stalled forks in response to PrimPol depletion is actually very small (from about 8% to 16% in Fig 2F). Doesn’t this indicate that most (84% at least) of the forks do not stall even when PrimPol is depleted and/or other pathways like TLS could predominate?

At the BPDE damage load induced under the treatment conditions, we estimate that only about 10% of all forks will encounter a lesion (see next point). This would fit with an additional 8% of forks being detected as stalled forks.

3. What would be the expected density of lesions caused by BPDE at this concentration? Do most forks never encounter a lesion? Would they generally only encounter a lesion on one of the template strands? A measurement of the actual lesion density if it is not known from the literature is needed to understand the results.

Based on our previous published measurements, we calculate that 10% of forks will encounter a lesion, which could be on either template strand). One fork encountering lesions on both strands would be rare. Specifically, treatment with 50 nM BPDE for 1 hour induces 400-500 adducts/10⁸

base pairs (PMID: 28593498; PMID: 24352536) so treatment with 50 nM BPDE for 20 min would induce 133-167 adducts/10⁸ base pairs. On average a replication fork replicates about 60 kb or 6x10⁴ base pairs (PMID: 17522385). There would be 0.08-0.1 BPDE adducts per 6x10⁴ base pairs.

We have included a summary of these calculations and references in the Results (line 147): “Based on our previous measurements^{26, 27}, we calculate that a 20 min treatment with 50 nM BPDE would induce 150 adducts/10⁸ base pairs. If an average replication fork replicates 6x10⁴ base pairs²⁸, about 10% of forks would likely encounter a lesion.”

4. The Vindigni lab reported that PrimPol function and expression are time-dependent in response to cisplatin lesions (Quinet 2020). Is this true for 4NQO?

Quinet et al. reported increased PrimPol mRNA and protein levels at 24h after both cisplatin and UV, although this seemed to be specific to BRCA1-deficient backgrounds. We could similarly use qRT-PCR and Western blotting for PrimPol at 24h after 4NQO. However, we argue that this experiment would not be necessary to support our conclusions. Firstly, the BRCA1-specific response to UV has already been reported. Secondly, we investigate HR responses to bulky lesions much earlier than 24h, and in BRCA1-proficient cells. Indeed, our existing qRT-PCR data (Figure S4B) show no increase in PRIMPOL mRNA expression over 24 hours after BPDE in WT U2OS cells.

We now display these data without the normalisation within each time point to emphasis this point, and added a note to the Discussion (line 376): “This work also showed that PrimPol mRNA and protein levels increase over 24 hours after cisplatin and UV treatment, thus favouring repriming over fork reversal as an adaptive mechanism⁴⁰. However, this response was specific for BRCA1-deficient backgrounds⁴⁰, and our qRT-PCR data show no increase in PRIMPOL mRNA expression over 24 hours after BPDE in BRCA1-proficient U2OS cells (Supplementary Figure 4B)”.

5. The continued RPA32 foci in RAD51 depleted cells after release from BPDE (Fig 2A) is interpreted as an involvement of RAD51 in resolution of the ssDNA. How do we know this is not because RAD51 is not available to replace RPA?

We would argue that these two scenarios are the same. If RAD51 would normally replace RPA, but is not able to due to siRNA depletion, this still supports that ssDNA would normally be bound by RAD51 in order to resolve it through HR.

6. More generally, the manuscript would be strengthened by additional data on resolution of damage and ssDNA gaps that formed. This is important since the authors are arguing that these gaps are eventually repaired via a post-replicative mechanism.

We considered trying to address this by performing a modified DNA fibre assay to monitor post-replicative gap filling (PMID: 28645379). This assay requires high doses of DNA damaging agent, so might not be feasible with 50 nM BPDE without substantial optimisation. Instead we would argue that we already show data on resolution of RPA foci, supporting that ssDNA is resolved (Fig. 2A). Our data further support that recombination promotes ssDNA repair, because RAD51 depletion attenuates RPA foci resolution (Fig. 2A). The low cytotoxicity of 50 nM BPDE and the PrimPol-dependent induction of sister chromatid exchanges also supports that damage is successfully repaired by PrimPol and homologous recombination.

We have now added additional data showing that γ H2AX signal is resolved with similar kinetics to RPA foci, further supporting that damage (likely ssDNA) is repaired (new Supplementary Fig. 2F). We added a description of these data to Results (line 192): “ γ H2AX foci were also resolved between 24 and 48 hours after release from BPDE, suggesting that they mostly mark ssDNA (Supplementary Figure 2F).”

7. It is striking that the authors do not examine markers of template switching or the requirements for these mechanisms in their analyses like PCNA ubiquitylation. Those analyses would strengthen the manuscript by adding additional mechanistic understanding of the pathway.

We would argue that our readouts for completed recombination, i.e. gene conversion in the recombination reporter and SCEs formation (Fig. 6A, C - E), are appropriate markers of template switching in yeast and vertebrate cells (e.g. PMID: 16783012; PMID: 12736307; PMID: 30487218). We have now re-written the Results section on HR and SCEs to make it clearer that we are measuring recombination events including template switching and cited the first two references above (line 327): "Both gene conversion (GC) and sister chromatid exchanges (SCEs) have been proposed as possible products of recombination by template switching^{36, 37}."

We are also very interested in examining the impact of PrimPol on PCNA ubiquitylation. Our preliminary data so far suggest that PCNA mono-ubiquitylation (TLS) might be reduced after PrimPol depletion, possibly reflecting presence of fewer ssDNA gaps. However, we still need to optimise the detection of PCNA poly-ubiquitylation, and fear that we may not be able to complete this if there are further lockdowns. We suggest that this would be work for a follow-up study. We have added to the Discussion (line 391): "Future work could involve examining the impact of PrimPol on PCNA ubiquitylation."

REVIEWERS' COMMENTS

Reviewer #2 (Remarks to the Author):

The authors have done a very thorough job in addressing the reviewers' concerns. I have only two very minor questions/comments that they might want to consider:

1. According to the authors' model, PrimPol supports fork progression through BPDE lesions by repriming. One would expect forks to slow down upon PrimPol depletion after BPDE treatment (i.e. shorter IdU tracts in siPrimPol compared to siControl in the second pulse after BPDE wash-off in their typical assay). Yet, this doesn't seem to be the case in Figure S3F (mock samples). The scenario seems to be different after UV, where replication slows down detectably upon PrimPol knockdown (Figure S6B). Would the authors speculate about a possible reason?

2. The authors explained in their rebuttal that "Based on the background numbers of foci that can be routinely observed in unchallenged U2OS cells; we find that they rarely have more than... 4 RAD51 foci." However, in Figure S1 the average number of foci per cell in unchallenged condition is around 20. Is this due to a very uneven distribution of foci, i.e. few cells with many foci and many cells without? In that case, it might be more informative to plot the exact number of foci per cell as a distribution in Figure S1D.

Reviewer #3 (Remarks to the Author):

I have no further comments for the authors. Thank you for clarifying the points in my review.

Response to reviewers' comments (Piberger et al.)

We thank the reviewers for their positive assessments.

Reviewer #2 (Remarks to the Author):

1. According to the authors' model, PrimPol supports fork progression through BPDE lesions by repriming. One would expect forks to slow down upon PrimPol depletion after BPDE treatment (i.e. shorter IdU tracts in siPrimPol compared to siControl in the second pulse after BPDE wash-off in their typical assay). Yet, this doesn't seem to be the case in Figure S3F (mock samples). The scenario seems to be different after UV, where replication slows down detectably upon PrimPol knockdown (Figure S6B). Would the authors speculate about a possible reason?

We agree that shorter IdU tracts in siPrimPol mock samples after BPDE would support our model, but we would not interpret too much into the data as the S1 fibre assay was not designed or optimised to detect fork stalling, unlike the protocol used e.g. in Figure 2F. We can only speculate that the S1 assay is not as sensitive for detecting the extra 10% fork stalling seen after BPDE. However, the CldU/IdU ratio is slightly increased in siPrimPol mock sample compared to nonT mock sample (Figure 2G) and this could reflect fork stalling. The overall larger effects seen after UV treatment could be due to higher DNA damage load, as the IdU shortening with S1 is also more pronounced in the UV experiments. We have added this statement and reference to the Results (line 296), which we hope will be helpful:

"UV-C irradiation induced large amounts of S1 endonuclease-sensitive sites, as the lesion density would be higher than after 50 nM BPDE at about 150 adducts/10⁶ base pairs³⁵."

2. The authors explained in their rebuttal that "Based on the background numbers of foci that can be routinely observed in unchallenged U2OS cells; we find that they rarely have more than... 4 RAD51 foci." However, in Figure S1 the average number of foci per cell in unchallenged condition is around 20. Is this due to a very uneven distribution of foci, i.e. few cells with many foci and many cells without? In that case, it might be more informative to plot the exact number of foci per cell as a distribution in Figure S1D.

We apologise if this was unclear. The quantification in Figure S1D is of average number of foci per foci-positive cell. Under solvent conditions, 15% of cells have more than 5 RAD51 foci and are scored foci-positive (Figure 1E). Most of these foci-positive cells have between 15 and 25 foci (see graphs below), considerably above the threshold, likely because they are experiencing some increased genotoxic stress even without BPDE treatment. It is true that there are few cells with many foci and many cells without. However, only cells with many foci were included in the analysis, so the graph in Figure S1D is not hiding an uneven distribution (see graphs below).

We could have set a threshold of 20 foci to exclude all of these background positive cells, but this would have not been very sensitive and might have hidden smaller increases in RAD51 foci, with implications for negative results such as with the higher BPDE concentrations as in Figure 1E.

We have amended the legend for Figure S1D to make it clearer: “Numbers of RAD51 foci per RAD51 foci-positive (> 5 foci) cells after release from 50 nM BPDE.”